# Actin filaments target the oligomeric maturation of the dynamin GTPase Drp1 to mitochondrial fission sites

**Wei-ke Ji[1†], Anna L Hatch[1†], Ronald A Merrill[2], Stefan Strack[2], Henry N Higgs[1*]**

[1]Department of Biochemistry, Geisel School of Medicine at Dartmouth, Hanover, United States; [2]Department of Pharmacology, The University of Iowa, Iowa City, United States

**Abstract** While the dynamin GTPase Drp1 plays a critical role during mitochondrial fission, mechanisms controlling its recruitment to fission sites are unclear. A current assumption is that cytosolic Drp1 is recruited directly to fission sites immediately prior to fission. Using live-cell microscopy, we find evidence for a different model, progressive maturation of Drp1 oligomers on mitochondria through incorporation of smaller mitochondrially-bound Drp1 units. Maturation of a stable Drp1 oligomer does not forcibly lead to fission. Drp1 oligomers also translocate directionally along mitochondria. Ionomycin, a calcium ionophore, causes rapid mitochondrial accumulation of actin filaments followed by Drp1 accumulation at the fission site, and increases fission rate. Inhibiting actin polymerization, myosin IIA, or the formin INF2 reduces both un-stimulated and ionomycin-induced Drp1 accumulation and mitochondrial fission. Actin filaments bind purified Drp1 and increase GTPase activity in a manner that is synergistic with the mitochondrial protein Mff, suggesting a role for direct Drp1/actin interaction. We propose that Drp1 is in dynamic equilibrium on mitochondria in a fission-independent manner, and that fission factors such as actin filaments target productive oligomerization to fission sites.

**\*For correspondence:** henry.higgs@dartmouth.edu

[†]These authors contributed equally to this work

**Competing interests:** The authors declare that no competing interests exist.

## Introduction

Mitochondrial fission and fusion create a dynamic network that is necessary for mitochondrial distribution and homeostasis. During mitosis, fission is necessary to distribute mitochondria between daughter cells. In neurons, fission and transport maintains mitochondrial distribution throughout the length of these highly polarized cells. Mitochondrial fission is also a fundamental step during mitophagy, a quality control mechanism that removes damaged mitochondrial segments. Dysregulated mitochondrial dynamics are linked to multiple neurological disorders, such as Alzheimer's, Huntington's, Parkinson's, hereditary ataxia, and Charcot-Marie Tooth disease (*Chen and Chan, 2009*; *Nunnari and Suomalainen, 2012*; *Girard et al., 2012*; *Niemann et al., 2005*).

The dynamin-like GTPase Drp1 is a central component of the fission machinery, with Drp1 oligomerization and constriction providing a driving force for the process. Drp1 is a cytosolic protein that translocates to the outer mitochondrial membrane (OMM) (*Smirnova et al., 2001*). In solution, Drp1 exists in a number of oligomeric states, including dimers, tetramers, and higher-order oligomers (*Fröhlich et al., 2013*; *Macdonald et al., 2014*). Membrane-bound oligomeric Drp1 induces tubulation of associated membrane, and constricts the membrane in the presence of GTP (*Francy et al., 2015*; *Mears et al., 2011*). Drp1 can bind to the OMM through several factors, including the mitochondrial-specific lipid cardiolipin (*Macdonald et al., 2014*; *Stepanyants et al., 2015*; *Bustillo-Zabalbeitia et al., 2014*; *Ugarte-Uribe et al., 2014*); and several 'receptors' on the OMM, such as Mff, MiD49, and MiD51 (*Richter et al., 2015*).

**eLife digest** Inside cells, structures called mitochondria supply the energy needed to carry out the processes that sustain life. Mitochondria constantly divide (a process known as fission) or fuse together, which helps to keep them in good working condition and well distributed around the cell. Several neurological disorders, including Parkinson's disease and Alzheimer's, are associated with problems that affect mitochondrial fission.

Many different molecules work together to help mitochondria divide, including a protein called Drp1. A number of Drp1 molecules can associate with each other to form an "oligomer" in the shape of a ring around a mitochondrion. The ring then constricts to split the mitochondrion in two.

It is often assumed that Drp1 molecules are recruited to the mitochondria immediately before fission and then form the oligomer ring. However, by using microscopy to track the movement of fluorescently labeled Drp1 molecules in human cells, Ji, Hatch et al. now suggest that Drp1 is continuously binding to and releasing from mitochondria, regardless of the need for fission. The experiments showed that when bound to surface of the mitochondrion, Drp1 switches between assembling and disassembling the oligomer ring. This process of Drp1 assembly and oligomerization on mitochondria is called maturation.

Specific signals for fission can push Drp1 toward maturation, which then leads to fission. Ji, Hatch et al. found that one such signal is the assembly of filaments of a protein called actin. Preventing actin filaments from forming reduced the amount of Drp1 that accumulated at mitochondria, and resulted in the mitochondria dividing less frequently. Further biochemical experiments also revealed that actin interacts directly with Drp1 and stimulates Drp1 activity, helping the ring to organize and assist mitochondrial fission.

The formation of actin filaments is not the only mechanism that can recruit Drp1 to mitochondria. Future work should investigate whether other mechanisms work with actin to recruit Drp1. As with actin filaments, other signals might be predicted to influence the balance of maturation and disassembly of Drp1 oligomers.

What mechanisms target specific sites on mitochondria for fission? Recent findings have shed light on this question. The endoplasmic reticulum (ER) forms close contacts with mitochondria at the eventual fission site (*Friedman et al., 2011*). Actin polymerization at the ER-mitochondrial interface may be a key component, controlled by the ER-bound formin INF2 (*Korobova et al., 2013*) and the mitochondrially-bound Spire1C protein (*Manor et al., 2015*). Myosin II activity also plays a role in this pathway (*DuBoff et al., 2012*; *Korobova et al., 2014*). Inhibition of actin polymerization, myosin II activity, or INF2 causes decreased levels of mitochondrially-bound Drp1 oligomers, suggesting that actin and myosin are acting at the level of Drp1 recruitment. Two additional actin binding proteins, cortactin and cofilin, have also been linked to mitochondrial fission (*Li et al., 2015*).

Given the number of factors that can mediate Drp1 recruitment to the OMM, as well as the necessity for Drp1 oligomerization at the fission site, a fundamental question concerns the relationship between Drp1 recruitment, oligomerization, and fission. We envision two possible models for the process, termed de novo assembly and targeted equilibrium. In de novo assembly, cytosolic Drp1 is recruited specifically to fission sites, where it oligomerizes and mediates fission in a concerted series of steps in response to fission signals (which include OMM receptors, cardiolipin and actin). In targeted equilibrium, Drp1 maintains a constant equilibrium between cytosolic and mitochondrial pools, as well as between dimer and oligomer. These equilibria are independent of fission. Fission signals shift these equilibria toward stable oligomerization at targeted fission sites on the OMM.

While few studies address the order of events during Drp1 recruitment, de novo assembly is often tacitly assumed (*Otera et al., 2013*; *Elgass et al., 2015*; *Labbé et al., 2014*). Here, we show evidence supporting targeted equilibrium. We observe that mitochondrially-bound Drp1 oligomers mature into stable oligomers through a series of merging events. Some of these Drp1 oligomers are motile—capable of translocating on the mitochondrial surface before engaging in fission. Notably, not all mitochondrially-bound Drp1 oligomers engage in fission. Using ionomycin treatment to induce mitochondrial fission, we show that actin polymerization precedes Drp1 oligomerization, and

that actin enriches with Drp1 at fission sites. Inhibition of actin polymerization, INF2 or myosin II decreases ionomycin-induced Drp1 accumulation and mitochondrial fission. Biochemically, actin filaments bind Drp1 directly and enhance its GTPase activity, suggesting that actin filaments can organize Drp1 into a productive oligomer. These results suggest that actin promotes fission by facilitating productive oligomerization of Drp1 on the OMM.

## Results

To monitor Drp1 dynamics in live cells, we developed a U2OS cell line stably expressing GFP-Drp1 with partial shRNA suppression of endogenous Drp1 (gDrp-U2OS). By quantitative western blotting, gDrp-U2OS cells maintain 1.64-fold total Drp1 levels compared to control U2OS cells, with 56% endogenous Drp1 and 44% GFP-Drp1 (*Figure 1—figure supplement 1A,B*). Mitochondrial morphology appears similar between control U2OS and gDrp-U2OS cells (not shown). We also developed a live-cell assay to measure mitochondrial fission rate, in which peripheral cellular regions are imaged over 10 min for dynamics of a fluorescent mitochondrial marker. Fission events are scored based on stable separation of the marker, and similar fission rates are obtained with either a matrix marker or an OMM marker (*Figure 1—figure supplement 1C*). Using this assay, the fission rates of control U2OS and gDrp1-U2OS cells are statistically similar (1.26 ± 1.01 versus 1.51 ± 0.69 fission events per mm mitochondrion per min, p = 0.49) (*Figure 1—figure supplement 1D*). Additionally, a second clone displaying undetectable endogenous Drp1 levels and GFP-Drp1 levels ~3.5-fold higher than control Drp1 levels displays similar mitochondrial fission frequency to control cells (*Figure 1*, *Figure 1—figure supplement 1E–G*), suggesting that GFP-Drp1 functionally compensates for endogenous Drp1 in these cells. We used the gDrp1-U2OS line for subsequent studies, due to its overall Drp1 levels being closer to those of control cells.

By live-cell confocal microscopy, much of the GFP-Drp1 in gDrp-U2OS cells appears diffuse in the cytoplasm (*Figure 1—figure supplement 2A*). However, adjusting the fluorescence threshold to eliminate this diffuse signal reveals abundant Drp1 'puncta' of varying intensity (*Figure 1A*, *Figure 1—figure supplement 2A,B*), likely representing Drp1 oligomers. We arbitrarily defined two categories of puncta: 'total' puncta, including all Drp1 signal over the diffuse background; and 'high-threshold' puncta, representing the brightest 30% of Drp1 puncta. These designations do not represent two distinct pools of intensity, as the distribution of punctum intensity is continuous (*Figure 1—figure supplement 2B*). Here, we use these categories to distinguish between smaller puncta and those approaching maximal size.

We employed a particle-tracking algorithm to analyze punctum position and lifetime relative to a mitochondrial matrix marker (*Figure 1C*). Approximately 49% of total puncta and 70% of high-threshold puncta co-localize with mitochondria at individual frames in the viewing period (*Figure 1B*). For most puncta, mitochondrial association is transient (*Video 1*), with mean lifetimes of 45.0 ± 104.8 s and 64.4 ± 113.1 s for total and high-threshold puncta, respectively (3582 and 1601 puncta analyzed). The largest population of puncta remains bound < 12 s in both cases (*Figure 1C*), although the high-threshold puncta are significantly more stably associated with mitochondria (p<0.001 for lifetime comparison). Puncta density on mitochondria does not vary systematically with mitochondrial length, with approximately 0.41 total puncta and 0.17 high-threshold puncta per μm mitochondrial length (*Figure 1—figure supplement 1H*).

How do Drp1 puncta assemble on mitochondria? We observed three distinct dynamic behaviors associated with mitochondrially-bound Drp1 puncta: punctum merging, punctum motility, and mitochondrial fission. Mitochondrially-bound puncta frequently merge together, with successive merging events resulting in punctum growth (*Figure 2A,B*, *Figure 2—figure supplement 1A*, *Video 2,3*). Merging is a reversible process, and can be followed by splitting of the merged punctum (*Figure 2—figure supplement 1B*, *Video 4*). The combination of these results suggests that a population of Drp1 is in rapid equilibrium between the cytosol and the OMM, and that one mechanism for assembly of larger Drp1 oligomers is through merging of mitochondrially-bound oligomers. We refer to this process as 'maturation'. Imaging at higher resolution by Airyscan microscopy suggests that maturing puncta are loosely associated with the mitochondrion, and progressively encircle the mitochondrion as they mature (*Figure 2C*, *Video 5*).

Low-intensity puncta are motile along the OMM, allowing them to interact during merging events (*Figure 2A,B*, *Figure 2—figure supplement 1A,B*). In addition, a sub-set of high-threshold Drp1

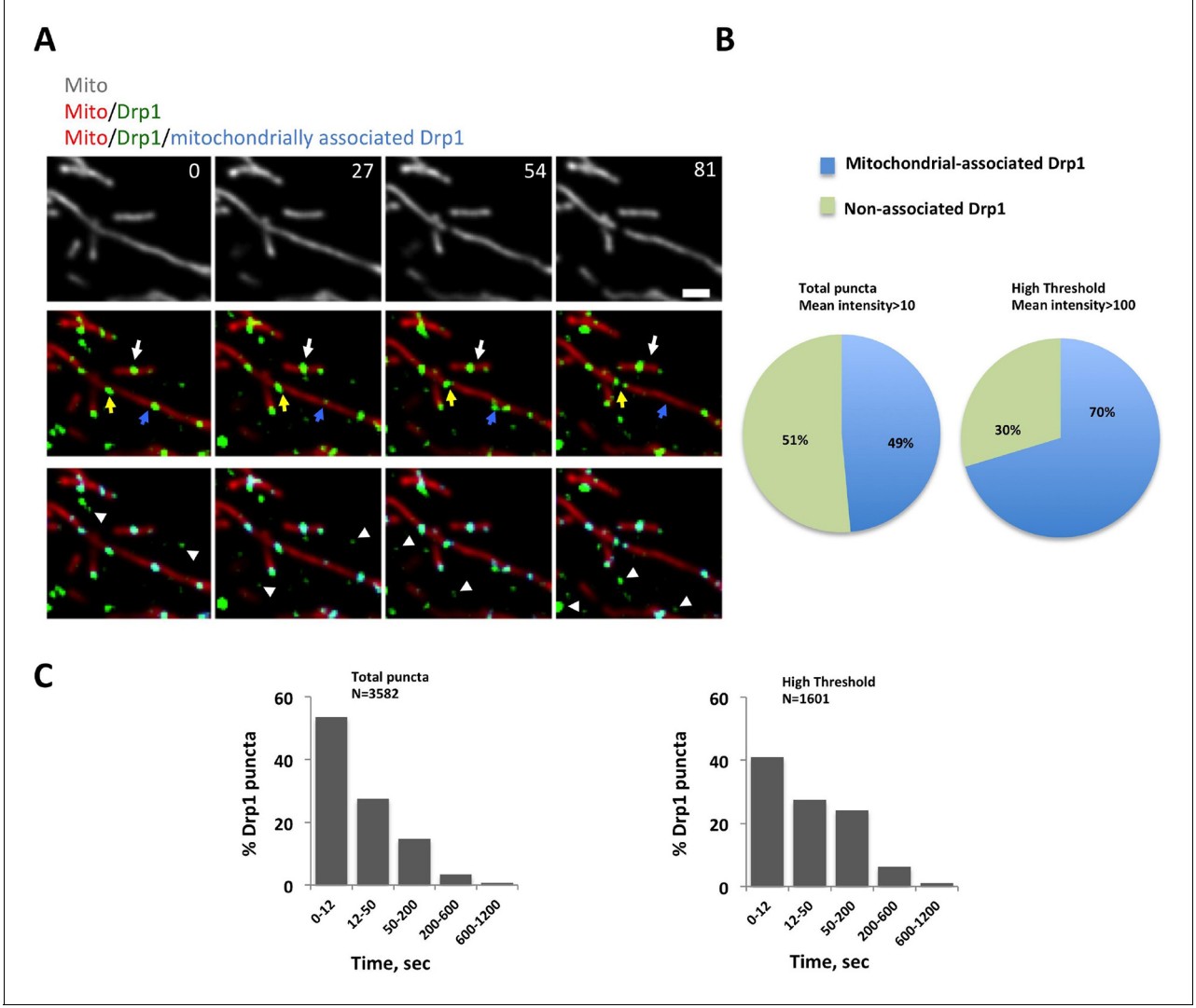

**Figure 1.** Mitochondrial association of Drp1 puncta. (**A**) Time-lapse image of region of gDrp1-U2OS cell transiently expressing mito-BFP and processed to remove background GFP (described in *Figure 1—figure supplement 2A*). Arrows denote variety of Drp1 puncta: stable (white arrow); transient (blue arrow); punctum at a fission site (yellow arrow). White arrowheads indicate unbound Drp1 puncta. Time in sec. Scale bar, 2 μm (*Video 1*). (**B**) Pie charts of mitochondria-associated versus non-associated Drp1 puncta: left panel, total puncta; right panel, high threshold. Each percentage is calculated by averaging percentages of mitochondrial Drp1 puncta over 201 frames from a 10 min imaging time. Quantification is based on 11 ROIs from 4 cells. (**C**) Histograms of lifetime distribution for mitochondrial Drp1 puncta, determined over 1200 s total recorded time. Left panel, total puncta; right panel, high threshold.

The following figure supplements are available for figure 1:

**Figure supplement 1.** Characterization of stable GFP-Drp1 cell line (gDrp1-U2OS cells).

**Figure supplement 2.** Image processing for GFP-Drp1 intensity in gDrp1-U2OS cells.

puncta translocates directionally along mitochondria for short distances at a mean velocity of 46.9 ± 12.8 nm/sec (*Figure 3A,B*). Assessing the fraction of motile puncta is challenging due to the underlying motility of the mitochondrion itself, thus our analysis is limited to situations where mitochondria are visibly stationary during the viewing period. Motile high-threshold puncta are capable of stopping abruptly and engaging in fission (*Figure 3A*, *Video 6*).

We next assessed the correlation between Drp1 puncta and mitochondrial fission, and found that all observed fission events are associated with high-threshold Drp1 puncta (49 events, examples in

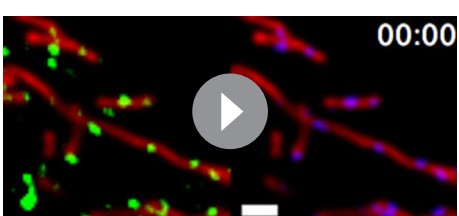

**Video 1.** Left: confocal time-lapse of Drp1 dynamics in gDrp1-U2OS cell transiently expressing mito-BFP; Right: mitochondrially-associated Drp1 puncta pseudo-colored in blue. Time lapse was taken in single z-plane every 3 s. Time min:sec. Bar, 2 µm (*Figure 1A*).

*Figure 1A*, *Figure 2A*, *Figure 2—figure supplement 1A*, *Figure 3A*). However, only a low percentage of Drp1 puncta (2.5% of total puncta and 6.3% of high threshold puncta) engages in observable fission events during the 10 min imaging period. Even for productive puncta, the lag between punctum establishment and mitochondrial fission varies significantly, with a mean of 76.0 ± 40.3 s (*Figure 3D*) that appears to be independent of mitochondrial length (*Figure 3E*). These results suggest that establishment of an apparently stable mitochondrially-bound Drp1 oligomer is not sufficient, and that additional steps are required to drive fission.

We made two additional observations concerning mitochondrial fission. First, mitochondria undergoing fission tend to be longer than the mean (*Figure 3—figure supplement 1A*). Second, Drp1 remains at newly created mitochondrial

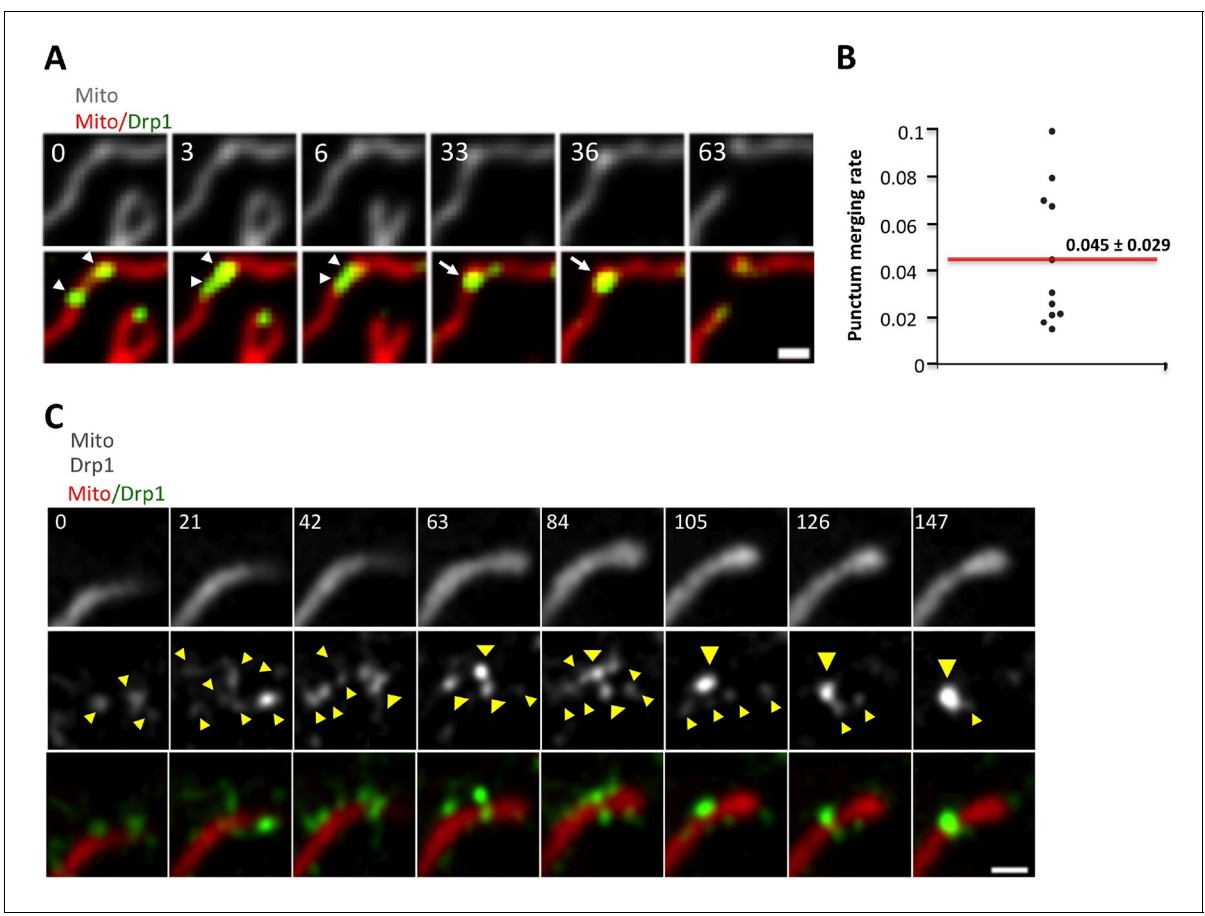

**Figure 2.** Maturation of mitochondrially-bound Drp1 puncta. (A) Example of Drp1 maturation events (arrowheads), followed by mitochondrial fission (63 s). Time in sec. Scale bar, 1 µm (*Video 2*). (B) Quantification of mitochondrial Drp1 merging rate, defined as number of merging events per min per number of Drp1 puncta. 5317 puncta from 11 ROIs in four cells based on Trakmate with parameters described before. Line indicates mean (0.045 ± 0.029) (C) Super-resolution Airyscan live-cell images showing Drp1 maturation (yellow arrowheads) (*Video 5*).

The following figure supplement is available for figure 2:

**Figure supplement 1.** Drp1 maturation events.

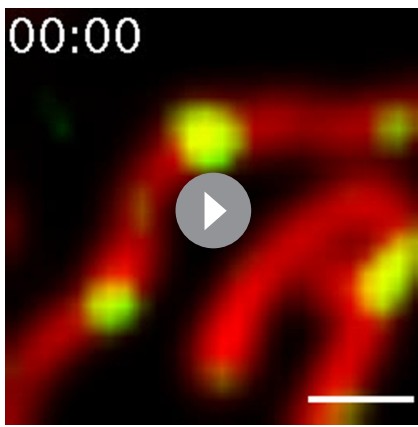

**Video 2.** Confocal time-lapse of Drp1 maturation in gDrp1-U2OS cell transiently expressing mCherry-mito7 (red). Time lapse was taken in single z-plane every 3 s. Time min:sec. Bar, 1 μm (*Figure 2A*).

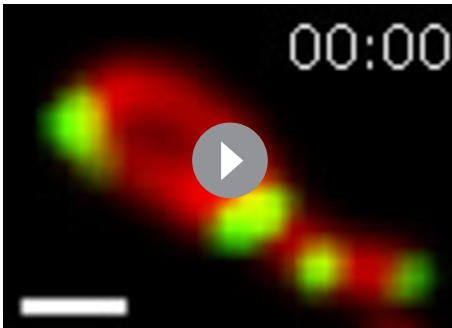

**Video 3.** confocal time-lapse of multiple Drp1 merging events, followed by fission at a looped junction in gDrp1-U2OS cell transiently expressing mCherry-mito7 (red). Time lapse was taken in single z-plane every 3 s. Time min:sec. Bar, 1 μm (*Figure 2—figure supplement 1A*).

ends for a significant time after fission. This residual Drp1 can be asymmetrically distributed, with one end inheriting most or all residual Drp1 (*Figure 3—figure supplement 1B–D*).

We examined Drp1 puncta at higher spatial resolution by live-cell 3D-structured illumination microscopy (3D-SIM), using Tom20-mCherry as the mitochondrial marker in order to label the OMM. We analyzed diameters of both mitochondria and associated Drp1 puncta for 'productive' events (leading to fission) and non-productive events (sites of stable Drp1 puncta that did not undergo fission during observation). The mean mitochondrial diameter in the absence of Drp1 is 322 ± 64 nm (*Figure 4B*). Non-productive high-threshold Drp1 puncta still cause significant constriction (*Figure 4—figure supplement 1A*), with mean mitochondrial diameter of 215 ± 50 nm (*Figure 4B*) and Drp1 diameter of 270 ± 38 nm (*Figure 4C*) at these sites. This constriction is maintained over time (*Figure 4D*). Productive Drp1 puncta undergo significant contraction immediately prior to fission, with mean Drp1 diameter of 200 ± 31 nm in the frame prior to fission (*Figure 4C,D*, *Figure 4—*

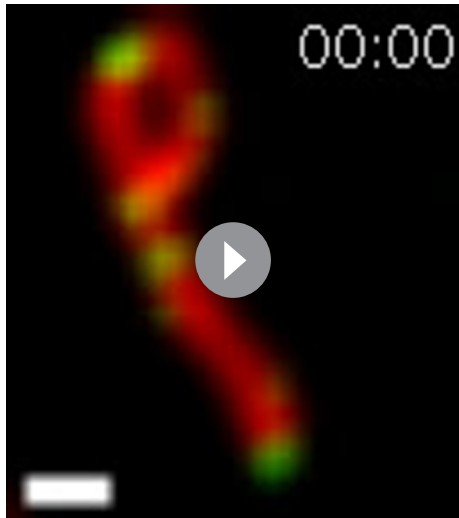

**Video 4.** Confocal live cell image of Drp1 merging event, followed by splitting of the punctum in gDrp1-U2OS cell transiently expressing mCherry-mito7 (red). Time lapse was taken in single z-plane every 3 s. Time min:sec. Bar, 1 μm (*Figure 2—figure supplement 1B*).

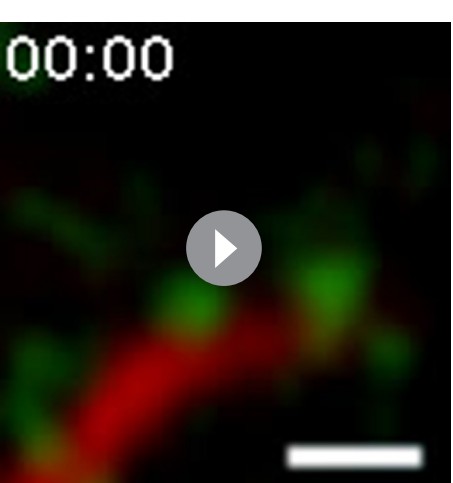

**Video 5.** Airyscan time-lapse of Drp1 maturation in gDrp1-U2OS cell (GFP in green) transiently expressing mCherry-mito-7 (red). Time lapse was taken in single z-plane in dorsal region of cells every 7 s. Time min:sec. Bar, 1 μm (*Figure 2C*).

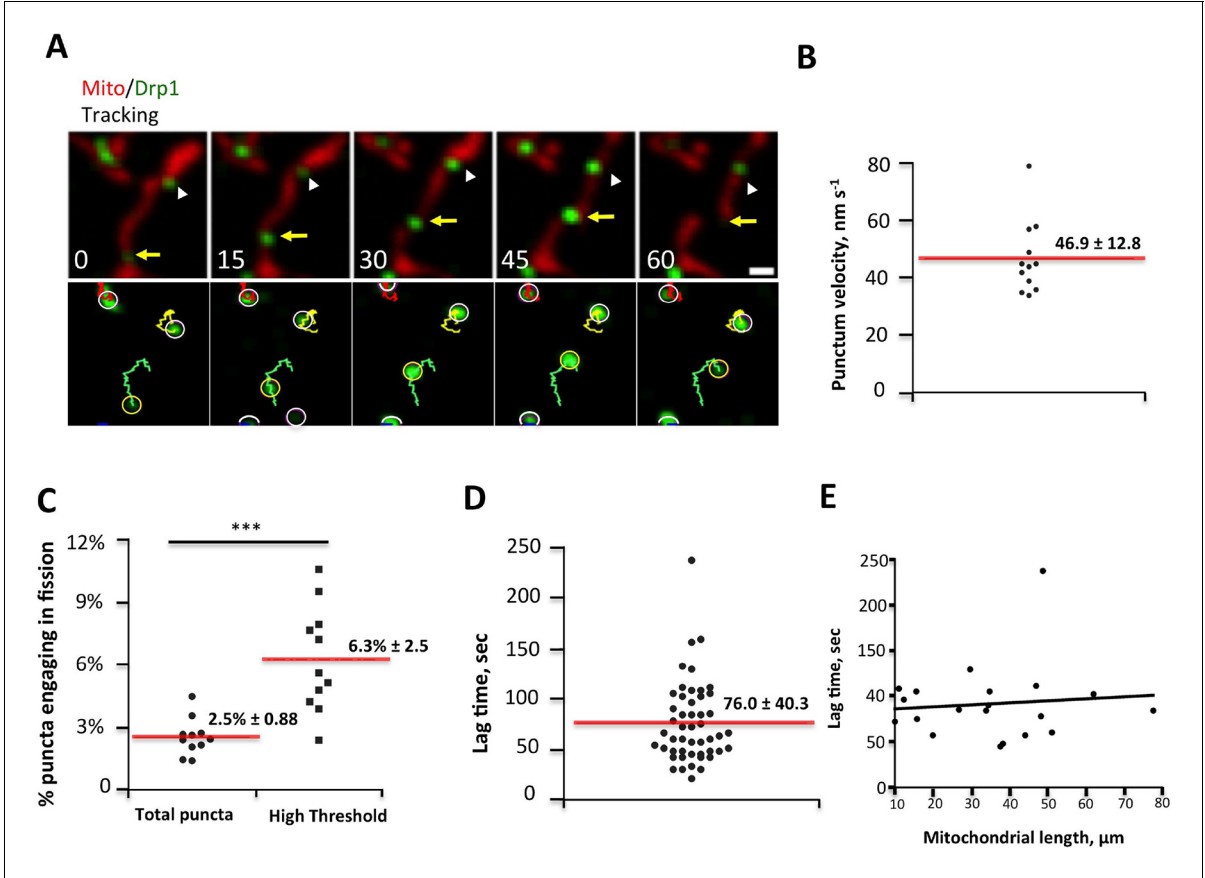

**Figure 3.** Drp1 puncta motility and relationship to mitochondrial fission. (**A**) Example of motile Drp1 punctum (yellow arrow) engaging in fission at 60 s, and a stationary punctum (white arrowhead). Lower panel maps tracks of these puncta. Time in sec. Scale bar, 1 µm (*Video 6*). (**B**) Drp1 punctum translocation velocity on mitochondrion (12 motile puncta from 12 ROIs from 11 cells). (**C**) Quantification of percentage of Drp1 puncta engaging in fission over the 10 min viewing period for total puncta and for high threshold puncta. 11 ROIs from 4 cells, \*\*\*p<0.001, unpaired Student t-test. (**D**) Lag time between Drp1 punctum appearance and fission for productive Drp1 puncta. 49 fission events from 11 cells. (**E**) Relationship between lag time and mitochondrial length for 20 mitochondria. Line fit data: y = 83.7 + 0.22x, RMSD = 0.096.

The following figure supplement is available for figure 3:

**Figure supplement 1.** Correlation between mitochondrial length and fission, and Drp1 punctum persistence on mitochondrial ends after mitochondrial fission.

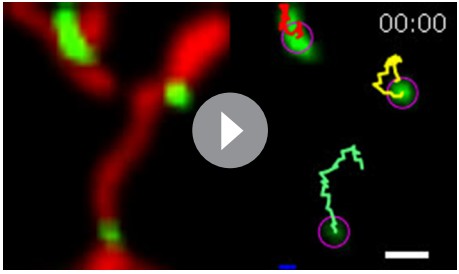

**Video 6.** Confocal time-lapse of Drp1 movement along mitochondrion in gDrp1-U2OS cell transiently expressing mito-BFP (red). Mitochondrially-bound Drp1 puncta were followed by Trackmate. Time lapse was taken in single z-plane every 1.5 s. Time min:sec. Bar, 1 µm (*Figure 3A*).

*figure supplement 1B*, *Video 7,8*). The mitochondrial diameter in this frame is 134 ± 25 nm, which is at the limit of resolution for 3D-SIM. Our results are qualitatively similar to those obtained from fixed-cell PALM studies (*Rosenbloom et al., 2014*), while adding the dynamic component of contraction prior to fission.

We also observed motile puncta by 3D-SIM. These puncta appear to distort as they translocate, periodically separating into strands (*Figure 5A*, *Figure 5—figure supplement 1A*, *Video 9,10*). Dynamic shape changes are not characteristic of stable puncta, which can be most easily appreciated when viewing a punctum transitioning from motile to stationary

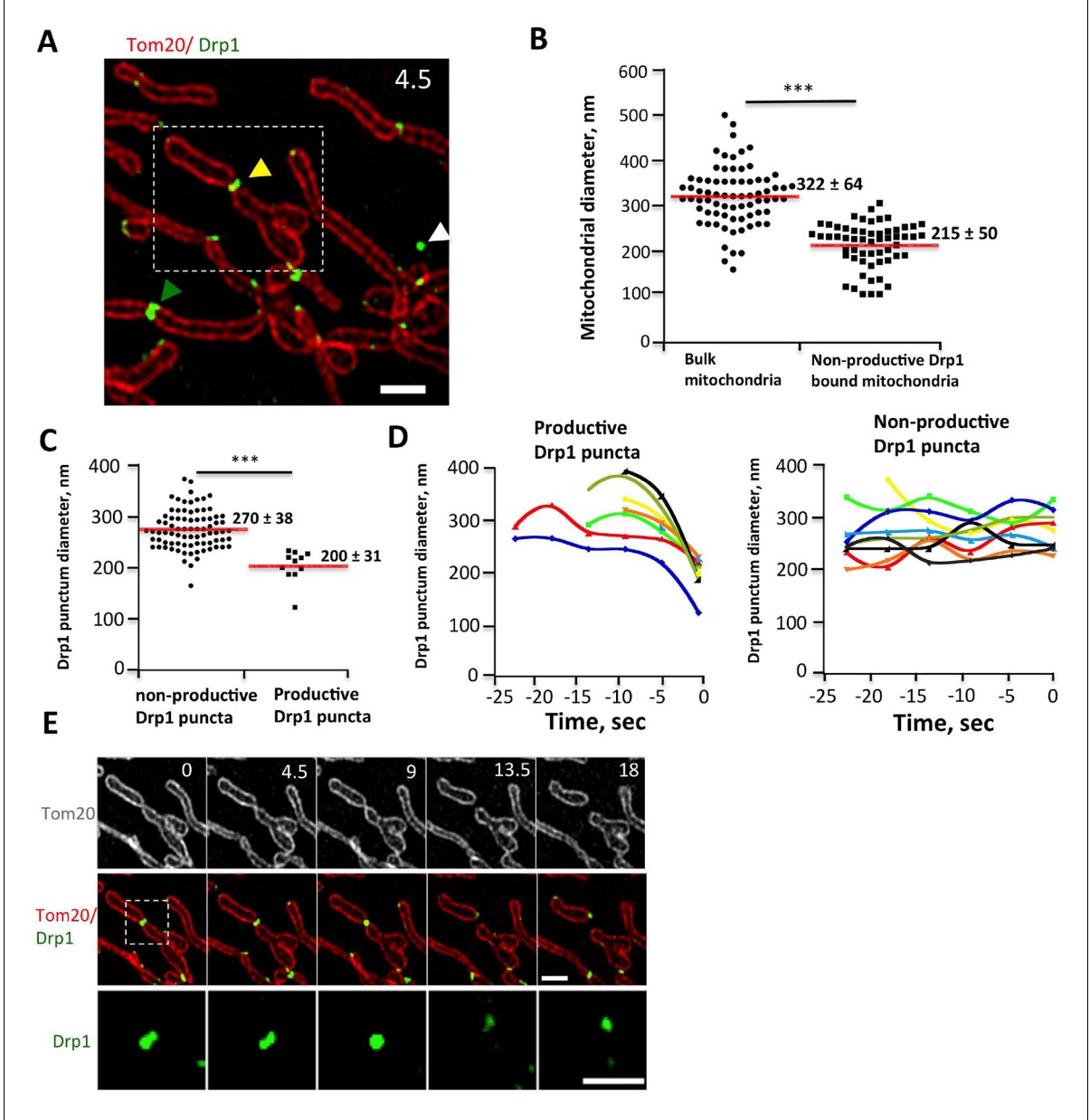

**Figure 4.** Drp1 and mitochondrial diameter by live-cell 3D-SIM. (**A**) 3D-SIM image of region of a live gDrp1-U2OS cell transiently expressing Tom20-mCherry. Yellow arrowhead, Drp1 punctum engaged in fission. Green arrowhead, non-productive punctum. White arrowhead, unbound punctum. Time in sec. Scale bar, 1 μm. (**B**) Quantification of mitochondrial diameters from regions devoid of Drp1 ("bulk mitochondria", left) or at sites of non-productive stationary puncta (right). ***p<0.001 unpaired Student t-test. (**C**) Quantification of Drp1 diameters for productive Drp1 puncta (4.5 s before fission, 11 events) versus non-productive puncta (82 events). ***p<0.001 unpaired Student t-test. (**D**) Diameter variation for seven productive Drp1 puncta over 30 s prior to fission (at 0 s, left) and for nine non-productive puncta over a similar time (right). (**E**) Time-lapse of fission event from (**A**) showing Drp1 punctum constriction. Bottom panel of Drp1 alone is further enlarged (***Video 7***). Time in sec. Scale bar, 1 μm (top); 0.5 μm (bottom).

The following figure supplement is available for figure 4:

**Figure supplement 1.** 3D-SIM imaging of Drp1 and mitochondria.

(***Figure 5B***). Motile puncta appear to encircle the mitochondrion to a significant degree, although the spatial resolution achieved here is not sufficient to answer this question definitively. Punctum velocity in 3D-SIM is similar to that measured by confocal microscopy (48.5 ± 17.1 nm/sec, ***Figure 5—figure supplement 1B***).

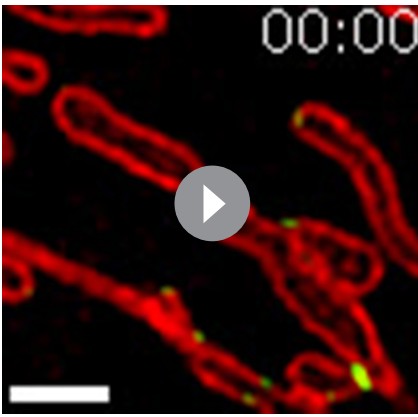

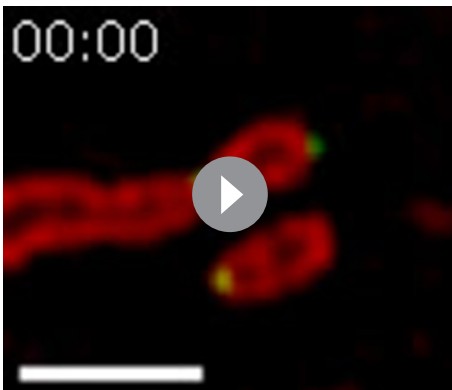

**Video 7.** 3D-SIM time-lapse of Drp1-mediated mitochondrial fission in gDrp1-U2OS cell transiently expressing Tom20-mCherry (red). Time lapse was taken every 4.5 s. Time min:sec. Bar, 1 μm (*Figure 4A,E*). DOI: 10.7554/eLife.11553.018

**Video 8.** 3D-SIM live cell image of Drp1-mediated mitochondrial fission in gDrp1-U2OS cell transiently expressing Tom20-mCherry (red). Time lapse was taken every 4.5 s. Time min:sec. Bar, 1 μm (*Figure 4—figure supplement 1B*). DOI: 10.7554/eLife.11553.019

What signal might stimulate Drp1 punctum maturation? We previously showed that mitochondrial length increased upon three cellular treatments: inhibition of actin polymerization by Latrunculin A (LatA), siRNA suppression of the formin protein INF2, or inhibition of myosin II by chemical inhibitors or siRNA (*Korobova et al., 2013*; *Korobova et al., 2014*). The increase in mitochondrial length suggests a fission defect, but could also reflect changes in mitochondrial fusion. Here, we establish that the effect is on mitochondrial fission, using our live-cell fission assay. Suppression of either INF2 or myosin IIA causes an ~50% decrease in fission rate, whereas Drp1 suppression or expression of the dominant-negative Drp1 K38A mutant cause near-complete inhibition (*Figure 6—figure*

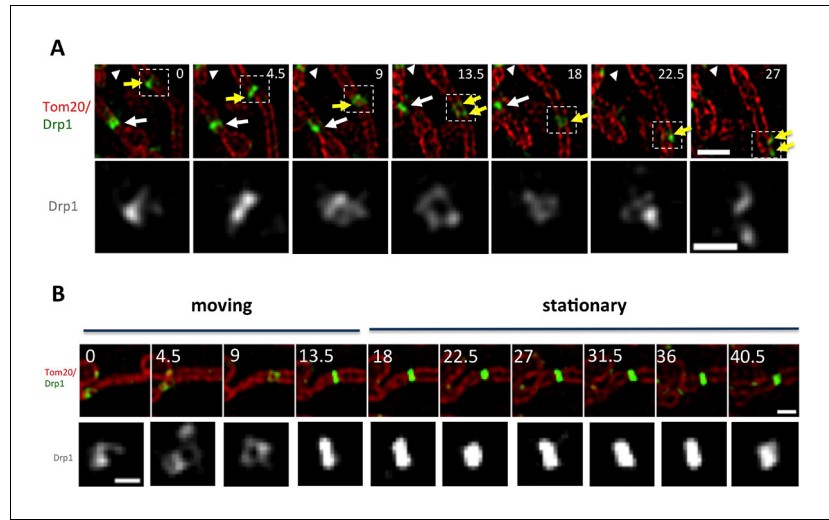

**Figure 5.** Drp1 motility by live-cell 3D-SIM. (**A**) Motile Drp1 punctum on a stationary mitochondrion (yellow arrow) and stationary Drp1 punctum on a motile mitochondrion (white arrow). White arrowhead indicates stationary Drp1 punctum on stationary mitochondrion. Lower panel is zoom of Drp1 alone, indicating changing Drp1 morphology during movement (*Video 9*). (**B**) Punctum transitioning from motile to stationary, accompanied by change in morphology. Time in sec. Scale bar, 1 μm in **A** (top); 0.5 μm in **A** (bottom); 1 μm in **B** (top); 0.3 μm in **B** (bottom). DOI: 10.7554/eLife.11553.020

The following figure supplement is available for figure 5:

**Figure supplement 1.** (A) Additional example of motile Drp1 punctum from 3D-SIM (similar to *Figure 5A*). DOI: 10.7554/eLife.11553.021

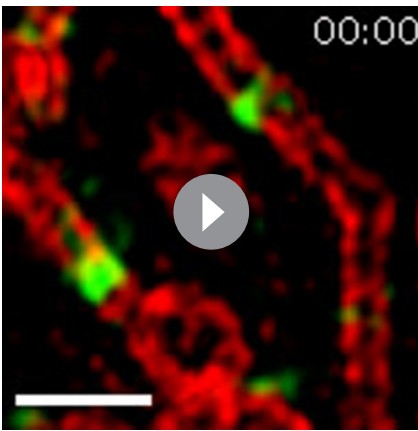

**Video 9.** 3D-SIM time-lapse of Drp1 movement along mitochondrion in stable gDrp1-U2OS cell transiently expressing Tom20-mCherry (red). Time lapse was taken every 4.5 s. Time min:sec. Bar, 1 μm (*Figure 5A*).

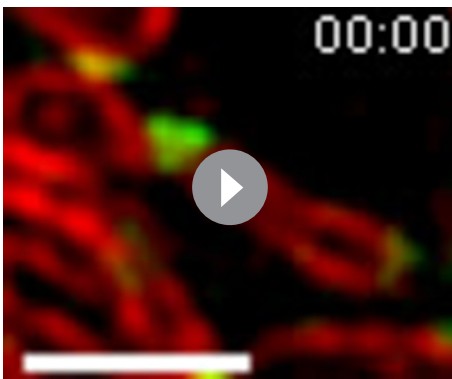

**Video 10.** 3D-SIM live cell image of Drp1 movement along mitochondrion in gDrp1-U2OS cell transiently expressing Tom20-mCherry (red). Time lapse was taken every 4.5 s. Time min:sec. Bar, 1 μm (*Figure 5—figure supplement 1A*).

*supplement 1A,B*). In addition, suppression of either INF2 or myosin IIA reduces mitochondrially-associated Drp1 puncta, with a moderate reduction of total Drp1 puncta (2.7-fold for both INF2 and myosin IIA) and a larger reduction in high threshold puncta (6.7-fold and 8.3-fold, respectively, *Figure 6—figure supplement 1C*). LatA treatment also causes a reduction in mitochondrially-associated Drp1 puncta, although the degree of high threshold reduction is not as great as for INF2 or myosin IIA suppression (*Figure 6—figure supplement 1D*). These results support the previous results suggesting a role for actin polymerization in mitochondrial Drp1 assembly.

We next examined the relationship between actin filaments and Drp1 during mitochondrial fission by live-cell microscopy in gDrp-U2OS cells. One challenge is the abundance of actin-based structures in U2OS cells. To mitigate this issue, we imaged cells at a focal plane significantly above the ventral surface, where there are fewer actin-based structures such as stress fibers. In this region, we observed frequent examples of actin filaments accumulating prior to fission (*Figure 6—figure supplement 2A*, *Video 11*), but the overall actin filament abundance in the region remained a challenge for determining whether its presence might contribute to fission. To assess the significance of the actin accumulation, we calculated the mean percentage of mitochondrial area covered by actin (*Figure 6—figure supplement 2B*), and compared this number to the percentage of fission events at which actin was present immediately prior to fission. We found that a significantly higher percentage of fission events (56%) were associated with actin filaments than would be expected by random correlation (28%, n = 59 fission events, *Figure 6— figure supplement 2C*).

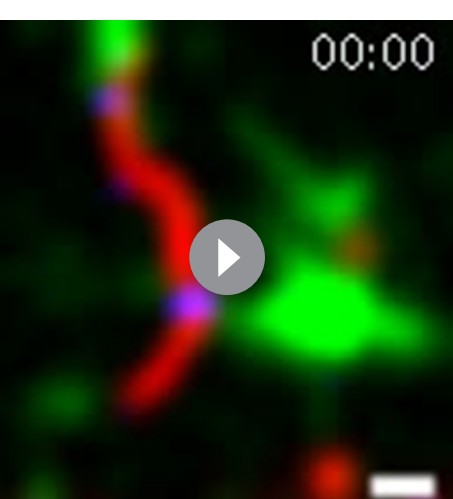

**Video 11.** Confocal live cell image of mitochondrial fission in an un-stimulated U2OS cell transiently expressing mApple-F-tractin and mito-BFP. Time lapse was taken in single z-plane in dorsal region of cells to avoid massive actin based structures every 3 s. Time min:sec. Bar, 1 μm (*Figure 6—figure supplement 1A*).

While this correlation is suggestive, the low frequency of fission events is challenging for temporal correlation of actin and Drp1 accumulation prior to fission. To examine this relationship in more detail, we used ionomycin, a calcium ionophore, to increase mitochondrial fission rate (*Tan et al., 2011*; *Sanmartin et al.,*

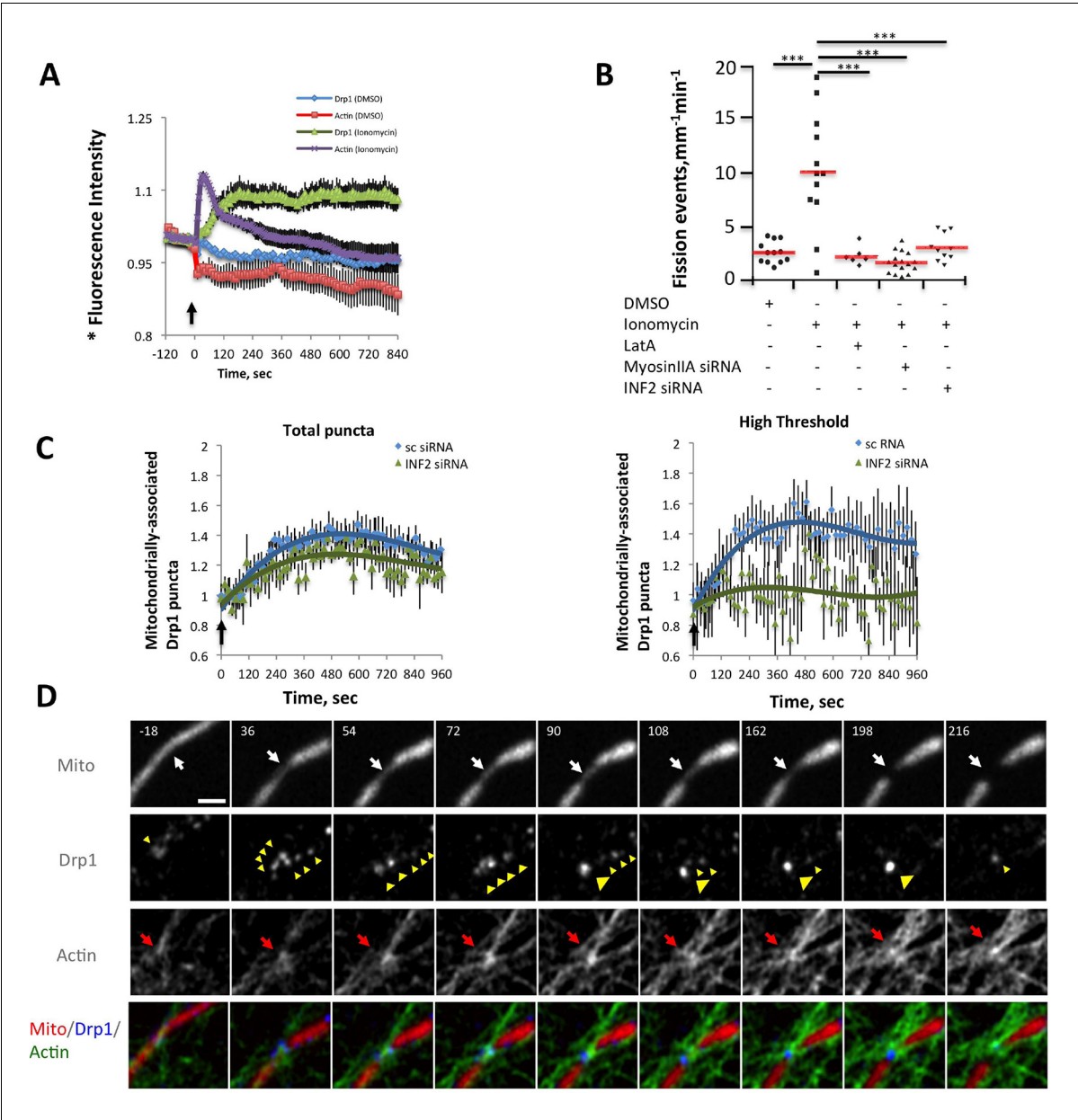

**Figure 6.** Ionomycin treatment induces actin polymerization, Drp1 maturation and mitochondrial fission. gDrp1-U2OS cells transiently transfected with mApple-F-tractin and mito-blue plasmids. (**A**) Time course of changes in Drp1 oligomer and actin filaments (judged by changes in GFP and mApple signal over cytosolic background, fluorescence normalized to time 0). GFP-Drp1 quantified over whole cell. Actin filaments were quantified from two or three ROIs per cell (approximately 3 × 3 μm each), in which no stress fibers or cell edges were included. DMSO (N = 10 cells) or ionomycin (4 μM, N = 12 cells) added at time 0 (arrow). * denotes total Drp1 puncta or polymerized actin fluorescence (as indicated for individual curves). (**B**) Mitochondrial fission rate (fission events per mm mitochondrial length per min) upon ionomycin treatment in the absence or presence of LatA (2 μM) or siRNA for INF2 or myosin IIA. \*\*\*p<0.001. (**C**) Quantification of mitochondrially bound Drp1 puncta (total puncta (left) and high threshold (right)) in response to ionomycin treatment in control (sc siRNA) and INF2 suppressed (INF2 siRNA) cells. Nine ROIs from six control cells and six INF2 suppressed cells. Ionomycin was added at 0 s to all samples (black arrow). Data first quantified as mitochondrially-bound puncta per mm mitochondrial length, then normalized such that 0 s value = 1. Error bars, standard deviation (S.D.). (**D**) Airyscan microscopy of a fission site (white arrow) showing actin filament enrichment (red arrows) and Drp1 maturation (yellow arrowheads) upon ionomycin treatment (1 μM at t = 0). Larger area shown in *Figure 6—figure supplement 2* (*Video 12*). Time in sec. Bar, 1 μm.

The following figure supplements are available for figure 6:

**Figure supplement 1.** Suppression of INF2 or myosin IIA decreases fission rate in un-stimulated conditions.

*Figure 6 continued*

**Figure supplement 2.** Actin filament enrichment with Drp1 puncta at fission sites in un-stimulated conditions.
**Figure supplement 3.** Comparison of mitochondrial fission rates in mitoRed labeled versus mitoBFP labeled cells.
**Figure supplement 4.** Larger field of super resolution Airyscan images, including the region shown in *Figure 6D*.

*2014*). A recent publication showed that 3T3 cells respond to increased cytosolic calcium with an acute burst of actin polymerization throughout the cytosol, mediated by INF2 (*Shao et al., 2015*). We find a similar ionomycin-induced actin burst in gDrp-U2OS cells (16.9 ± 8.7% signal increase, $t_{1/2}$ 28.1 ± 19.8 s) followed by a rapid decline (*Figure 6A*). Ionomycin treatment also increases mitochondrial fission rate 3.9-fold (2.61 ± 1.01 vs 10.23 ± 5.41 events $mm^{-1}$ $min^{-1}$) (*Figure 6B*). Similar fission rate increases occur when using mitochondrial matrix markers tagged with either dsRed or BFP (*Figure 6—figure supplement 3*).

We next asked whether the quantity of Drp1 oligomers also increased following ionomycin treatment, initially monitoring total Drp1 puncta. Indeed, ionomycin treatment causes Drp1 oligomer signal increases from 10-20%, depending on the experiment, with a consistent $t_{1/2}$ of 99.9 ± 19.0 s (*Figure 6A*, also see *Figure 7A,C and D*). In contrast to the transient actin increase, the Drp1 oligomer signal remains elevated for >10 min. These measurements are rapid to conduct, allowing analysis of every time point in the sequence, but do not provide information on the change in mitochondrially-bound Drp1 puncta. For this reason, we also quantified the change in the number of mitochondrially-associated Drp1 puncta upon ionomycin treatment on a more limited set of time points, and found ~40% increase for low- or high-threshold puncta (*Figure 6C*).

In addition, we visually monitored the dynamics of actin and Drp1 at individual ionomycin-induced fission sites by Airyscan microscopy. Prior to ionomycin treatment, a number of low intensity Drp1 puncta dynamically associate with mitochondria, and a meshwork of actin exists throughout the cytoplasm (*Figure 6D*, *Figure 6—figure supplement 4*, *Video 12*). This actin meshwork enriches at certain positions on mitochondria, and Drp1 puncta tend to be more abundant at these sites but are dynamic. Upon ionomycin treatment, the actin signal intensifies throughout the cytosol, particularly at sites of mitochondrial crossover. Drp1 puncta mature at these actin-enriched sites, and some of these sites undergo fission while others undergo constriction but no fission during the viewing period (*Figure 6D*, *Figure 6—figure supplement 4*, *Video 12*).

We tested the association between actin and Drp1 further by inhibiting actin polymerization, followed by ionomycin stimulation. Pre-treatment for 15 min with LatA abolishes both Drp1 accumulation (*Figure 7A,B*) and the increase in mitochondrial fission (*Figure 6B*). Suppression of INF2 by siRNA treatment potently inhibits the actin increase, but causes a smaller decrease in whole-cell Drp1 oligomer accumulation than that achieved by LatA (*Figure 7C*). However, the effect of INF2 suppression on mitochondrially-associated high threshold Drp1 puncta is dramatic, with almost complete elimination of the ionomycin-induced increase (*Figure 6C* Right). INF2 suppression also causes a decrease in ionomycin-induced mitochondrial fission (*Figure 6B*). Interestingly, INF2 suppression has a negligible effect on Drp1 puncta of lower intensity (*Figure 6C* Left). These results suggest that INF2-mediated actin polymerization plays a role in Drp1 maturation on mitochondria.

We next tested the role of myosin II in this process. Suppression of myosin IIA does not alter the actin response to ionomycin (signal intensity increase 20.0 ± 6.4% with $t_{1/2}$ of 22.8 ± 12.7 s), but strongly reduces Drp1 oligomerization (*Figure 7D*) and mitochondrial fission (*Figure 6B*). These experiments suggest that both INF2-induced actin polymerization and myosin II activity are important pre-requisites for productive Drp1 oligomer assembly.

The mechanism by which actin might stimulate Drp1 accumulation is unclear. One possibility is that Drp1 binds directly to actin filaments at the fission site, similar to the direct binding between dynamin and actin (*Gu et al., 2010*; *Palmer et al., 2015*; *Mooren et al., 2009*). Interestingly, a previous study demonstrates that actin accumulates at clathrin-mediated endocytosis sites prior to dynamin 2 arrival (*Grassart et al., 2014*). We tested the possibility of a direct interaction using purified proteins. Through co-sedimentation assay, Drp1 binds actin with an apparent $K_d$ of 0.8 ± 0.4 μM (*Figure 8A*, *Figure 8— figure supplement 1A*). Interestingly, less than 50% of the Drp1 binds actin

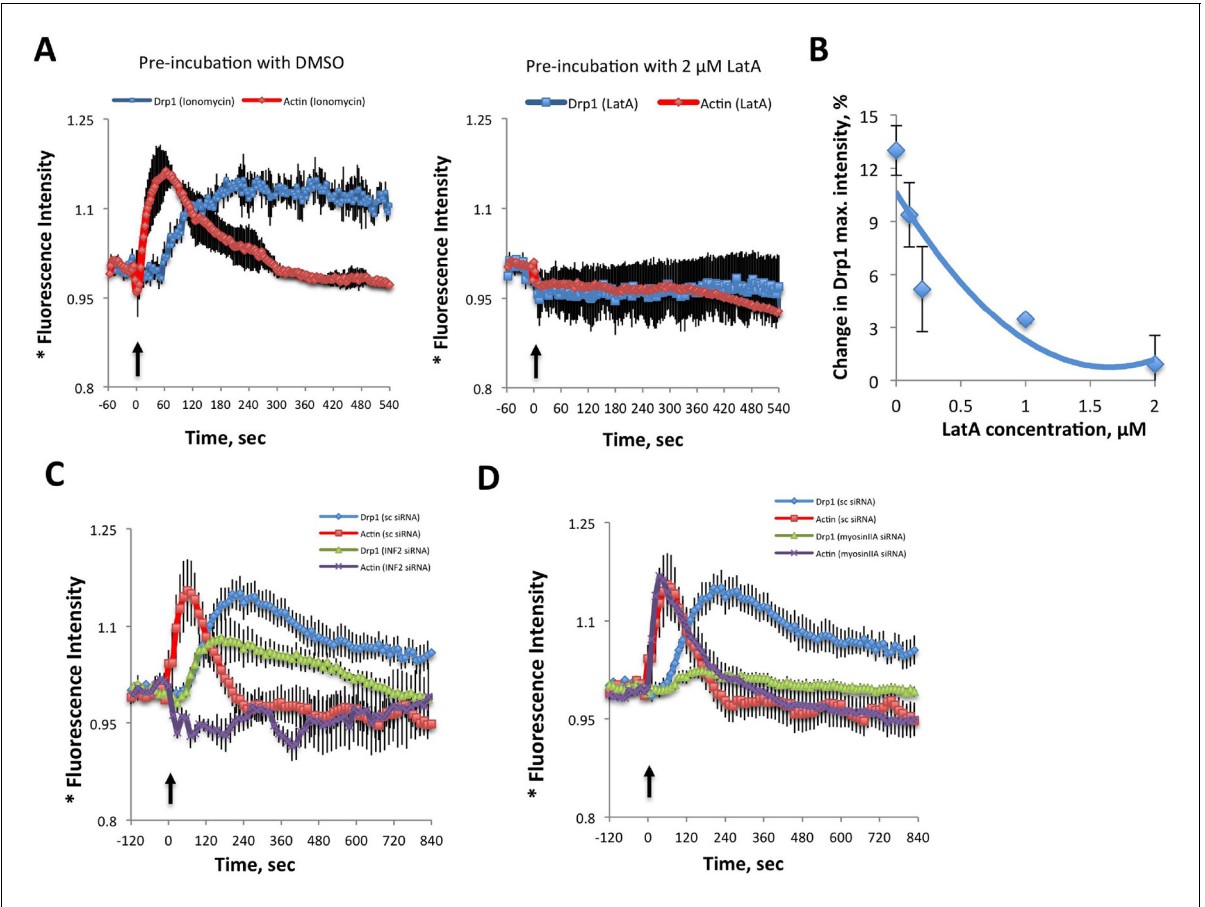

**Figure 7.** Inhibition of actin, INF2 or myosin IIA reduces ionomycin-induced Drp1 maturation. (A,B) gDrp1-U2OS cells were transiently transfected with mitochondrial matrix marker, mito-BFP, and actin filament marker, mApple-F-tractin. Cells were treated with 0, 0.1, 0.2, 1 and 2 µM LatA (or DMSO) for 15 min before imaging. At 60 s after starting imaging, cells were treated with 4 µM ionomycin (in the presence of the appropriate concentration of LatA) and imaged for 9 min. GFP-Drp1 signals over cytosolic background (background subtract, ImageJ) were measured per whole cell; actin filament signals were quantified from two or three ROIs per cell (approximately 3 × 3 µm), in which no stress fibers or cell edges were included. Error bars, standard deviation for **A** and S.E.M. for **B**. (**C**) Time course of ionomycin-induced changes in Drp1 oligomer and actin filaments after INF2 siRNA treatment (INF2 siRNA). 11 control cells (sc siRNA), six INF2 siRNA. Error bars, S.E.M. (**D**) Time course of ionomycin-induced changes in Drp1 oligomer and actin filaments after myosin IIA siRNA treatment. N = 11 control cells (sc siRNA), 12 myosin IIA siRNA. * denotes total Drp1 puncta or polymerized actin (as indicated for individual curves). Error bars, S.E.M for **C** and **D**.

filaments even at saturating actin concentrations (*Figure 8A*). This situation is not caused by a significant pool of 'inactive' Drp1 in our purified preparation, since >90% of the Drp1 pellets when incubated with a non-hydrolyzable GTP analogue (*Figure 8—figure supplement 1B*). We postulate that partial binding is due to differential affinities of Drp1 oligomeric states for actin filaments. In these assays, Drp1 does not change the critical concentration of actin, since actin polymerizes to a similar extent in the absence or presence of Drp1 (*Figure 8—figure supplement 1C*).

By total internal reflection (TIRF) microscopy, GFP-tagged Drp1 binds evenly throughout the actin filament (*Figure 8B*), with no apparent preference for specific filament regions (*Figure 8C*, *Video 13*). By negative-stain

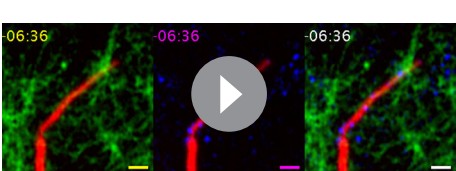

**Video 12.** Airyscan time-lapse of mitochondrial fission in response to Ionomycin treatment in gDrp1-U2OS cell (GFP in blue) transiently expressing mApple-F-tractin (green) and mito-BFP (red). Time lapse was taken in single z-plane in dorsal region of cells every 18 s. Time min:sec. Bar, 1 µm (*Figure 6D*).

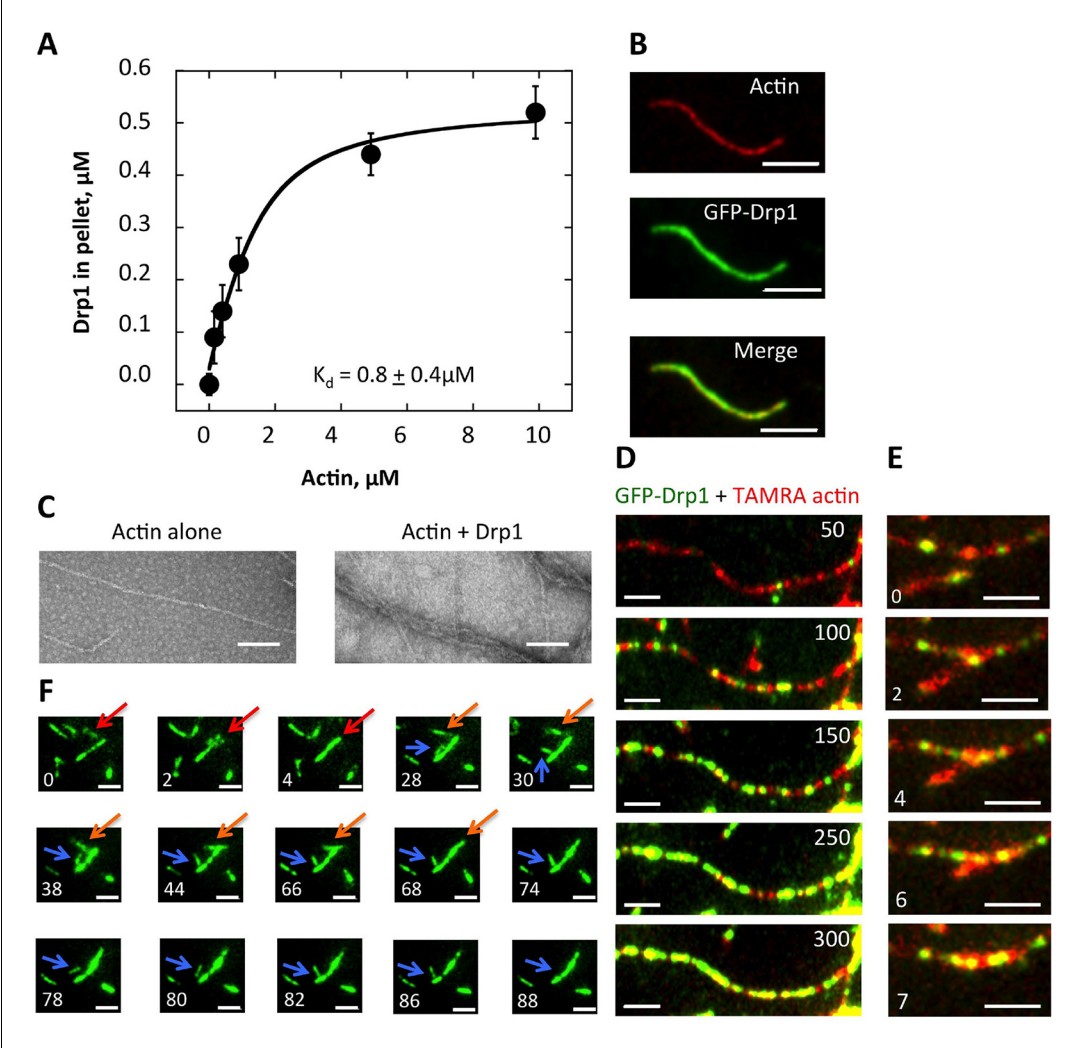

**Figure 8.** Drp1 binds to actin filaments. (**A**) Co-sedimentation assay in which Drp1 (1.3 µM) is incubated with indicated concentration of pre-polymerized actin (concentrations indicate total actin) for 1 hr, then centrifuged at >100,000 ×g to sediment actin filaments. Pellets analyzed by SDS-PAGE (*Figure 8—figure supplement 1A*). Each data point is the mean from 10 independent experiments. Error bars, standard deviation. (**B**) Single time point images from TIRF microscopy assay of actin filaments (20% TAMRA-labeled) mixed with saturating concentration of GFP-Drp1. Scale bar, 2 µm. (**C**) Negative stain electron microscopy of 2 µM actin filaments in the absence or presence of 1 µM Drp1. Mean filament widths: 8.9 + 0.2 nm (n = 44 filaments) for actin alone; and 27.2 + 1.3 nm (n = 49) for actin/Drp1. Scale bar, 50 nm. (**D**) TIRF microscopy time-lapse montage showing GFP-Drp1 dynamics on an actin filament. Time indicates seconds after GFP-Drp1 addition. Scale bar, 2 µm (*Video 13*). (**E**) TIRF microscopy time-lapse montage showing GFP-Drp1 can bundle actin filaments (20% TAMRA-labeled). Time indicated in seconds. Scale bar, 2 µm (*Video 14*). (**F**) TIRF microscopy time-lapse montage showing multiple bundling events by GFP-Drp1 (denoted by red, orange and blue arrows). Actin filaments not shown. Note Drp1-coated filament denoted by blue arrow, which binds by its end to a second filament for ~50 s, before releasing, flipping, binding by its opposite end, then bundling into the second filament. Time in sec. Scale bar, 2 µm (*Video 15*).

The following figure supplement is available for figure 8:

**Figure supplement 1.** Binding of Drp1 to actin filaments by co-sedimentation assay.

electron microscopy, Drp1-bound filaments have mean widths of 27.2 ± 1.3 nm, compared to 8.9 ± 0.2 nm for actin alone (*Figure 8D*). It is unclear whether this increased width is due to Drp1 binding a single filament or to Drp1-mediated filament bundling. The width of a Drp1 dimer is 22.9 nm (*Fröhlich et al., 2013*), similar to our measured width. However, Drp1 can induce filament bundling, as evidenced by TIRF microscopy (*Figure 8E*, *Video 14*). Indeed, time-lapse imaging of the bundling process suggests that filaments have a preferred bundled orientation, and can be end-bound to the sides of Drp1-bound filaments prior to bundling (*Figure 8F*, *Video 15*).

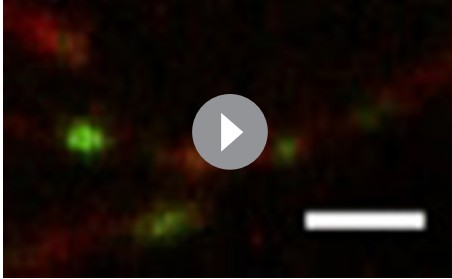

**Video 13.** TIRF microscopy time-lapse GFP-Drp1 (270 nM) was added to TAMRA-actin filaments (1 $\mu$M, 20% TAMRA initially). Actin filaments were polymerized for 10 min prior to GFP-Drp1 addition. Dual-color simultaneous images were collected every 1 s. Scale bar, 2 $\mu$m. 307 frames played at 75 ms frame rate (13.3-fold accelerated) (*Figure 8C*).

**Video 14.** TIRF microscopy time-lapse showing filament bundling. GFP-Drp1 (270 nM) was added to TAMRA-actin filaments (1 µM, 20% TAMRA initially). Actin filaments were polymerized for 10 min prior to GFP-Drp1 addition. Dual-color simultaneous images were collected every 1 s. Scale bar, 2 µm. 13 frames played at 250 ms frame rate (fourfold accelerated) (*Figure 8E*).

Drp1 binding to anionic lipids such as cardiolipin increases its GTPase activity (*Macdonald et al., 2014*), presumably by placing its GTPase domains in close proximity as is found for other dynamin family proteins (*Bui and Shaw, 2013*). We tested whether actin filaments could also increase Drp1's GTPase activity and found a ~3.5-fold increase (*Figure 9A*). An additional question is whether actin filaments can synergize with Drp1 receptors on the OMM. We tested this possibility using the cytosolic portion of Mff. Mff alone causes only a slight increase in Drp1 GTPase activity at the concentrations tested. However, the combination of Mff and actin filaments causes a substantial increase in Drp1 activity, far beyond the additive effects of either Mff or actin alone (*Figure 9B*). These results show that Drp1 binds actin filaments in a manner that stimulates its catalytic activity, and that actin filaments can synergize with Mff to increase productive Drp1 oligomerization.

## Discussion

Overall, our results support several important and novel features of Drp1 oligomerization. Rather than being an 'all or none' process in which Drp1 oligomerization at the fission site is induced de novo by fission signals, we postulate that both Drp1 oligomerization and mitochondrial association are in constant equilibrium even in the absence of fission signals. Fission signals serve to 'target' this equilibrium to fission sites. Actin filaments are one such fission signal.

We illustrate these features as a three-stage model for fission-productive Drp1: recruitment, maturation, and conversion (*Figure 10*). In recruitment, 'units' of Drp1 are in rapid equilibrium between cytosolic and mitochondrially-bound pools. These Drp1 units diffuse on the OMM, and merge to form larger units. Maturation is the progressive association of mitochondrially-bound Drp1 units, leading to assembly of an oligomer that fully encircles the mitochondrion. The size of the mature oligomer is unclear, but might be similar to the 26-40 subunits found for dynamin2 (*Grassart et al., 2014*; *Cocucci et al., 2014*). Mature oligomers are capable of movement along the OMM. Conversion represents a distinct process in which a site

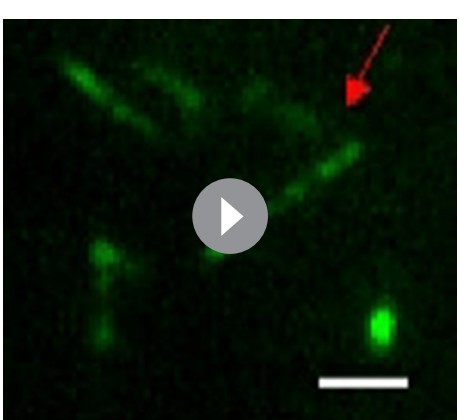

**Video 15.** TIRF microscopy time-lapse showing filament bundling. GFP-Drp1 (2 µM) was added to TAMRA-actin filaments (1 µM, 20% TAMRA initially). Actin filaments were polymerized for 10 min prior to GFP-Drp1 addition. Images were collected every 2 s. Scale bar, 2 µm. 46 framed played at 100 ms frame rate (20-fold accelerated). Red, yellow and blue arrows indicate bundling events (*Figure 8F*).

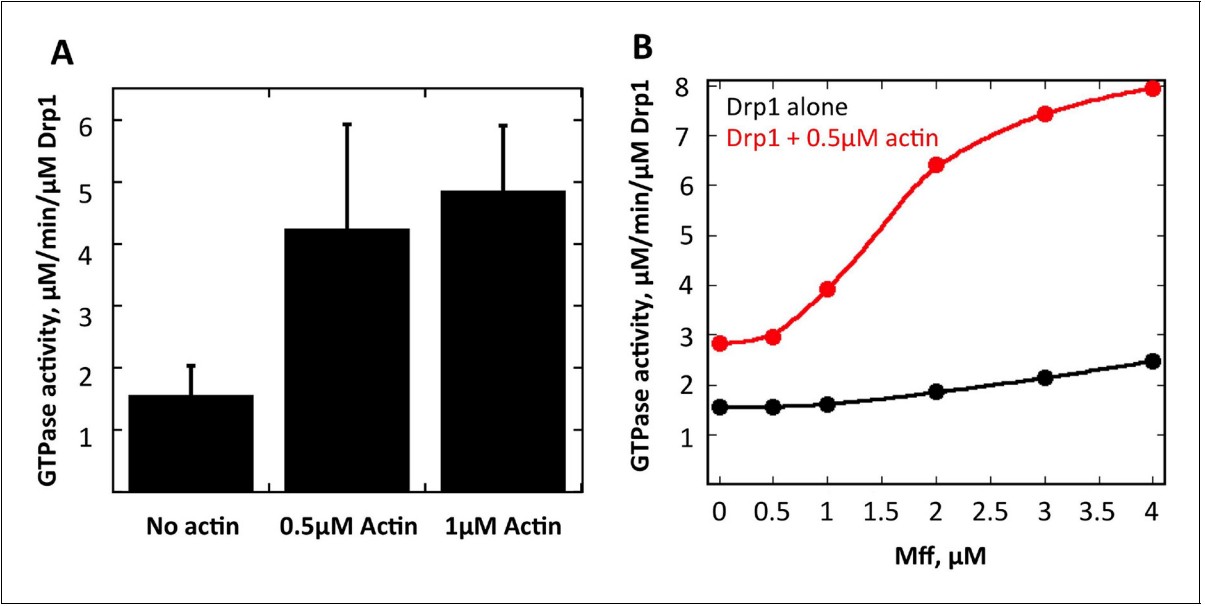

**Figure 9.** Actin fialments stimulate Drp1 GTP hydrolysis synergistically with Mff. (**A**) GTPase assays containing 1 µM Drp1 in the presence or absence of 0.5 or 1 µM actin (pre-polymerized for 1 hr) for 5 min before GTP addition (250 µM). N = 6 experiments. (**B**) GTPase assays containing 1 µM Drp1 in the presence or absence of 0.5 µM actin (pre-polymerized for 1 hr) and the indicated concentration of Mff (cytosolic region) for 5 min before GTP addition (250 µM).

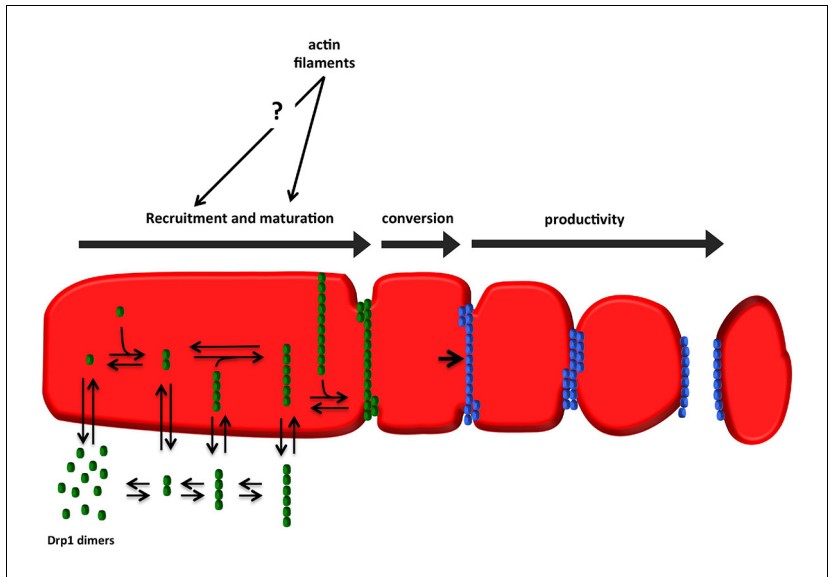

**Figure 10.** Model for assembly of fission-productive Drp1 on mitochondria. **Step 1: recruitment**. Drp1 units are in equilibrium between cytosol and OMM, possibly binding to OMM receptors such as Mff, MiD49, MiD51, or Fis1, or to cardiolipin. We suggest that several distinct oligomeric species may interact with the OMM. **Step 2: maturation**. Mitochondrially-bound Drp1 oligomers grow through incorporation of other mitochondrially-bound oligomers, progressively encircling the OMM in the process. The larger Drp1 oligomers constrict the OMM but are not yet competent to drive fission. **Step 3: conversion**. A stable Drp1 oligomer becomes productive for mitochondrial fission (change from green to blue), with increased Drp1 ring constriction driving membrane ingression. Actin filaments and myosin II clearly stimulate the maturation process, with possible effects on recruitment as well.

containing a stable Drp1 oligomer is rendered fission-competent.

This model raises many new testable questions. First, what is the complete list of "fission signals"? There is evidence that OMM proteins such as Mff, MiD49/51, and Fis1, as well as cardiolipin exposure on the OMM, can serve as fission signals. We postulate that actin filaments in proximity to the mitochondrion are another such signal. Other potential factors that might influence fission could include mitochondrial length and mitochondrial branches. We find that fission events occur disproportionately on longer mitochondria, but we do not find an increase in Drp1 puncta density on longer mitochondria, suggesting that maturation is not favored on long mitochondria. It should be realized, however, that most mitochondria exist in a branched network in these cells. The relationship between mitochondrial branching and mitochondrial fission is a fascinating question still to be addressed.

Second, do the various fission signals act in series or in parallel? In other words, do the signals work together within the same fission pathway, or act in alternate pathways? Both options could be true, depending on the fission signal. For example, we postulate that actin filaments and cardiolipin might represent alternate maturation factors, since both are polyvalent anions. In contrast, actin filaments could work in concert with Mff, either in "coincidence detection" by Drp1 (*Hatch et al., 2014*) or by actin filaments serving as a reservoir for delivery of Drp1 oligomers to Mff. Recent work has shown that Mff binds preferentially to oligomerized Drp1 (*Liu and Chan, 2015*), which is consistent with this possibility. We show here that actin filaments and Mff synergize in stimulating Drp1's GTPase activity. A final possibility is that specific fission factors mediate distinct steps in the process. For example, a recent study on the membrane-altering ability of the Drp1-cardiolipin interaction might suggest that cardiolipin triggers a conversion step (*Stepanyants et al., 2015*).

Third, what is the size of the Drp1 'unit' that is recruited to mitochondria? We cannot detect the smallest oligomers (dimers, tetramers) in this study for several reasons, including: our threshold procedure forcibly removes these from analysis, and only ~ half of the Drp1 in these cells is GFP-labeled. The available data suggest that the recruited unit may be mixed oligomers of GFP-labeled and endogenous Drp1. Recent biochemical studies have shown that Drp1 exists in equilibrium between oligomeric states in solution, with some evidence that dimers are the relevant membrane-recruited unit in the case of cardiolipin binding (*Fröhlich et al., 2013*; *Macdonald et al., 2014*). On the other hand, in the present study we observe clear instances of Drp1 puncta apparently translocating from cytosol to mitochondrion. We state 'apparently' because it is possible that these oligomers transfer from another membrane or a cytoskeletal structure. Based on these findings, we postulate that a range of oligomeric species can be recruited to mitochondria, but that their on- and off-rates might vary considerably.

Fourth, what steps are stimulated by actin? Our data show that inhibiting actin polymerization inhibits the accumulation of Drp1 oligomers, suggestive of roles in either recruitment or maturation. Direct Drp1 binding to actin filaments near mitochondria might enhance initial mitochondrial binding, or might accelerate oligomerization steps of cytosolic Drp1 prior to mitochondrial interaction, both of which would aid recruitment. Alternately, actin binding to mitochondrially-bound Drp1 might enhance oligomerization on the mitochondrial surface, which would represent maturation. The two possibilities are not mutually exclusive. A related question is: what is myosin II's role in the process? One possibility is that myosin II activity might lead to Drp1-independent 'pre-constriction' of the OMM, as we have proposed previously (*Korobova et al., 2014*; *Hatch et al., 2014*). It is also possible that myosin II organizes the INF2-assembled actin filaments in a manner optimal for Drp1 recruitment and/or maturation.

Fifth, what are the roles of other actin binding proteins that have been identified as contributing to mitochondrial fission, such as cortactin, cofilin, and Spire 1C (*Manor et al., 2015*; *Li et al., 2015*)? There is evidence that mitochondrially-bound Spire 1C and ER-bound INF2 work through the same pathway, suggesting that Spire 1C might serve as a nucleation factor and INF2 as an elongation factor and/or a severing protein. Spire proteins cooperate with other formin proteins in a similar manner (*Vizcarra et al., 2011*; *Quinlan, 2013*). Roles for cortactin and cofilin in the same pathway as Spire 1C/INF2 are possible, but an alternative is that they contribute to a distinct actin-dependent pathway. The fact that steady-state fission rate is only partially reduced by INF2 or myosin IIA suppression (*Figure 6—figure supplement 1*) suggests some degree of functional redundancy.

Sixth, does the conversion step represent a change in Drp1 structure/activity, or changes in other fission components? One possibility is recruitment of additional necessary molecules. For example,

recent publications show that endophilin co-operates with actin and dynamin in clathrin-independent endocytosis (*Boucrot et al., 2015*; *Renard et al., 2015*). Alternately, exposure of cardiolipin on the OMM could trigger a conversion step, as mentioned above (*Macdonald et al., 2014*).

Seventh, how does ionomycin stimulate mitochondrial fission? We postulate that, in addition to the known effect of calcium on Drp1 phosphorylation state (*Cribbs and Strack, 2007*), increased cytosolic calcium activates INF2, as demonstrated in recent work (*Shao et al., 2015*) (R. Wedlich-Söldner, personal communication). These effects are likely to be similar to those induced by the Listeria protein LLO (*Stavru et al., 2013*), which is also a calcium ionophore.

Eighth, what is the role of the ER in mitochondrial fission? The tight association of ER with mitochondria at fission sites (*Friedman et al., 2011*) could contribute in two ways. First, ER supplies the INF2 isoform responsible for actin dynamics at the fission site (*Korobova et al., 2013*; *Manor et al., 2015*). Second, ER might contribute to the calcium dynamics necessary for ionomycin-mediated fission, considering the complex relationship between extracellular calcium entry, calcium release from ER, and ER-mitochondrial calcium communication (*Horne and Meyer, 1997*; *Csordás et al., 1999*).

Finally, what is the mechanism of Drp1 motility on mitochondria? Our 3D-SIM time-lapse movies suggest that motile Drp1 oligomers are rings that encircle the mitochondrion, and that these rings change significantly during the motility process, in a manner similar to 'walking' along the mitochondrial surface. Another possible mechanism could be 'treadmilling' of Drp1 units, adding to one side of the Drp1 oligomer while dissociating from the other side, akin to cytoskeletal polymers. Furthermore, what is the purpose of this motility? One possibility is something we call the 'night watchman' hypothesis, in which motility allows mature Drp1 oligomers to 'patrol' the mitochondrion in search of fission signals.

## Materials and methods

### Plasmids and siRNA oligonucleotides

Mito-DsRed and mito-BFP constructs were previously described (*Korobova et al., 2014*), and consist of amino acids 1–22 of S. cerevisiae COX4 N-terminal to the respective fusion protein. mCherry-mito-7 was purchased from Addgene (#55102), and consists of the mitochondrial targeting sequence was from subunit VIII of human cytochrome C oxidase N-terminal to mCherry. Tom20-mCherry was a gift from Andrew G. York (NIH, Bethesda, MD) and described in *York et al. (2013)*. Drp1-containing plasmids (GFP (A206K)-Drp1 and GFP (A206K)-Drp1 K38A) were described in *Strack et al. (2013)*. These plasmids co-express H1 promoter–driven shRNA for endogenous Drp1 as well as GFP-tagged, RNAi-resistant rat Drp1. mApple-F-tractin plasmid was a gift from Clare Waterman and Ana Pasapera (NIH, Bethesda, MD), and described in (*Johnson and Schell, 2009*). Human Mff isoform 8 cDNA (*Gandre-Babbe and van der Bliek, 2008*) was amplified through RT-PCR from Hela cell RNA. Mff protein containing amino acids 1-197 (NP_001263994) and Ala-2 Cys substitution, for protein labeling, was cloned into the pET21a (Novagen) vector using the NdeI and XhoI sites.

Oligonucleotides for human total INF2 siRNA were synthesized by IDT Oligo against target sequence 5′- GGAUCAACCUGGAGAUCAUCCGC-3′ (siRNA#1), and 5′- GCAGUACCGCUUCAGCA-UUGUCA-3′ (siRNA#2). Oligonucleotides for INF2 CAAX isoform were 5′-ACAAAGAAACTGTGTGT-GA-3′ (siRNA#1), and 5′- CCCTGATTCTGATGATAAT-3′ (siRNA#2). Oligonucleotides for human Drp1siRNA were synthesized by IDT Oligo against target sequence 5′-GCCAGCUAGAUAUUAAC-AACAAGAA-3′ (siRNA#1) and 5′- GGAACGCAGAGCAGCGGAAAGAGCT-3′ (siRNA#2). Oligonucleotides for human myosin IIA siRNA were synthesized by IDT Oligo against target sequence 5′-GCCACGCCCAGAAGAACGAGAAUGC-3′ (siRNA#1) and 5′- GCAAGCUGCCGAUAAGU-AUCUCUAT-3′ (siRNA#2). As a control, Silencer Negative Control 5′-CGUUAAUCGCGUAUAAU-ACGCGUAT-3′ (Ambion) was used.

### Cell culture, transfection and drug treatment

Human osteosarcoma U2OS (obtained directly from American Type Culture Collection (HTB-96) in 2014) were grown in DMEM (Invitrogen, Carlsbad, CA, USA) supplemented with 10% calf serum (Atlanta Biologicals). The U2OS line was tested at regular intervals for mycoplasma contamination using LookOut Mycoplasma PCR detection kit (Sigma-Aldrich). U2OS cells are not on the list of mis-identified or cross-contaminated cell lines compiled by the International Cell Line Authentication

Committee (ICLAC), and we have not had them verified by a third party. The stable GFP-Drp1 U2OS cell line (gDrp1-U2OS). The gDrp1-U2OS cell line was made by transient transfection of the GFP (A206K)-Drp1 plasmid into U2OS cells, followed by selection in G418. Selected cells were then flow-sorted for GFP signal as single cells onto 96-well plates, and individual clones were analyzed for GFP-Drp1 expression and endogenous Drp1 suppression. Cell lines we used for a maximum of 20 passages.

For transfection of the U2OS or gDrp1-U2OS lines, cells were seeded at $4 \times 10^5$ cells per well of a 6-well dish ~16 hr prior to transfection. Plasmid transfections were performed in OPTI-MEM media (Invitrogen) with 2 μL Lipofectamine 2000 (Invitrogen) per well for 6 hr, followed by trypsinization and re-plating onto concanavalin A (ConA, Sigma/Aldrich, Cat. No. C5275)- coated coverslips (25 mm, Electron microscopy Sciences, Cat. No.#72225-01), at ~$3.5 \times 10^5$ cells per well. Cells were imaged in live cell media (Cat.No. 21063-029, Life technologies), ~16–24 hr after transfection.

For all experiments, the following amounts of DNA were transfected per well (individually or combined for co-transfection): 400 ng for mito-BFP; 300 ng for GFP-Drp1 and GFP-Drp1 K38A constructs; 850 ng for Tom20-mCherry; 900 ng for mCherry-mito7; 500 ng for mApple-F-Tractin. For siRNA transfections, cells were plated on 6 well plates with 30–40% density, and 2 μl RNAimax (Invitrogen) and 63 pg of siRNA were used per well. Cells were analyzed 72–84 hr post-transfection for suppression.

For Ionomycin treatment, gDrp1-U2OS cells were transfected with mitoBFP and mApple-F-tractin, as described above the day before imaging. Cells were mounted on the microscope for imaging, then treated with 4 μM Ionomycin (from a 4 mM stock in DMSO, Sigma/Aldrich, Cat. No. I6034) at 20 frames (1 or 2 min, depends on time interval used) during imaging. Medium was pre-equilibrated for temp and $CO_2$ content before use. DMSO was used as the negative control.

For Latrunculin A (LatA) pre-incubation followed by ionomycin treatment, cells were transfected with mitoBFP and mApple-F-tractin the day before treatment. Cells were incubated with live cell medium containing LatA (added from a 0.2 mM DMSO stock) for 15 min before imaging, and ionomycin was added at frame 20 (1 or 2 min depends on time intervals) with DMSO used as the negative control.

## Live imaging and confocal microscopy

Cells were grown on 25 mm coverslips coated with ConA (coverslips treated for ~2 hr with 100 μg/mL ConA in water at room temperature). Coverslips were mounted into flow chambers, then onto a Wave FX spinning disk confocal microscope (Quorum Technologies, Inc., Guelph, Canada, on a Nikon Eclipse Ti microscope), equipped with Hamamatsu ImageM EM CCD cameras and Bionomic Controller (20/20 Technology, Inc) temperature-controlled stage set to 37°C. After equilibrating to temperature for 10 min, cells were imaged with the 60x 1.4 NA Plan Apo objective (Nikon) using the 403 nm and 450/50 filter for BFP, 491 nm laser and 525/20 filter for GFP, and the 561 nm laser and 593/40 filter for mApple or mCherry.

## Live cell super-resolution 3D-SIM microscopy

Super-resolution 3D-SIM images were acquired on a DeltaVision OMX V4 (GE Healthcare) equipped with a 60x/1.42 NA PlanApo oil immersion objective (Olympus), 405, 488, 568 and 642 nm solid state lasers (100 mW) and sCMOS cameras (pco.edge). Image stacks of 1 μm with 0.125 μm thick z-sections and 15 images per optical slice (3 angles and 5 phases) with were acquired using immersion oil with a refractive index 1.524. Images were reconstructed using Wiener filter settings of 0.005 and optical transfer functions (OTFs) measured specifically for each channel with SoftWoRx 6.1.3 (GE Healthcare) to obtain super-resolution images with a twofold increase in resolution both axially and laterally. Images from different color channels were registered using parameters generated from a gold grid registration slide (GE Healthcare) and SoftWoRx 6.1.3 (GE Healthcare).

## Live-cell super resolution Airyscan microscopy

Super resolution images were acquired on LSM 880 equipped with 63x/1.4 NA plan Apochromat oil objective, using the Airyscan detector (Carl Zeiss Microscopy, Thornwood, NY). The Airyscan uses a 32-channel array of GaAsP detectors configured as 0.2 Airy Units per channel to collect the data that is subsequently processed using the Zen2 software. The processing involves an online reassignment

of the pixel information followed by linear deconvolution. The end result is a 1.7-fold improvement in resolution in X, Y and Z and at least fourfold improvement in signal to noise ratio.

## Image analysis for mitochondrially associated Drp1 puncta

gDrp1-U2OS cells transiently transfected with mitochondrial markers were imaged live by spin disc confocal fluorescence microscopy every 3 s for 10 min in a single focal plane. Regions of interest with readily resolvable mitochondria and Drp1 were processed as described in *Figure 1—figure supplement 2A*. We thresholded mitochondrially associated Drp1 puncta by using a ImageJ plugin, Colocalization, with the following parameters: Ratio 50%(0–100%); Threshold channel 1: 30 (0–255); Threshold channel 2: 30 (0–255); Display value: 255 (0–255).

Mitochondrially associated Drp1 puncta were further analyzed by Trackmate V2.7.3 (as described in *Figure 1—figure supplement 2B,C*) to separate into total puncta and high threshold categories. Parameters used in Trackmate are: estimated blob diameter of 0.7 microns for confocal and 0.5 microns for SIM; LoG detector settings: tracker – LAP tracker, frame to frame linking of 1 micron, track segment gap-closing of 1 micron max distance and 1 frame max frame gap, track segment splitting of 1 micron, and track segment merging of 1 micron. The number of Drp1 puncta in each category were automatically counted frame-by-frame by ImageJ macro, Find Stack Maxima.

## Quantification of mitochondrial length and fission rate

Suitable ROI's were selected for analysis based on whether individual mitochondria were resolvable and did not leave the focal plane. Files of these ROIs were assembled, then coded and scrambled by one investigator, and analyzed for fission by a second investigator in a blinded manner as to the treatment condition. The second investigator scanned the ROIs frame-by-frame manually for fission events, and determined mitochondrial length within the ROI using the ImageJ macro, Mitochondrial Morphology (described in *Dagda et al. (2009)*). The results were then given back to the first investigator for de-coding. For measuring lengths of individual mitochondria, ROIs were selected that enabled imaging of the entire mitochondrial length where possible. Due to the fact that the majority of mitochondrial mass is in the form of a branched mitochondrial 'network' in these cells, and that one end of the network is often in the peri-nuclear region which is difficult to resolve, it was frequently difficult to find both ends of the network, in which case the resolvable length was reported.

## Antibodies

Polyclonal antibodies against human INF2 N-terminus (amino acids 1-424) or FH1-FH2- C (amino acids 469-1249, CAAX) were raised in rabbits by Covance (Denver, PA), and affinity purified using DID construct (amino acids 1-269) or FH1-FH2 (amino acids 469-940) coupled to Sulfolink (Thermo/Pierce). Anti-Tubulin (DM1-$\alpha$, Sigma/Aldrich) was used at 1:10,000 dilution. Drp1 was detected using a rabbit monoclonal antibody (Cell Signaling) at 1:500 dilution. MyosinII-A was detected using a rabbit polyclonal antibody (Cell Signaling) at 1:500 dilution.

## Western blotting

For Western Blotting, cells were grown on 6 well plate, trypsinized, washed with PBS and resuspended 50 µL PBS. 50 µL was mixed with 34 µL of 10% SDS and 1 µL of 1 M DTT, boiled 5 min, cooled to 23°C, then 17 µl of 300 mM of freshly made NEM in water was added. Just before SDS-PAGE, the protein sample was mixed 1:1 with 2xDB (250 mM Tris-HCl pH 6.8, 2 mM EDTA, 20% glycerol, 0.8% SDS, 0.02% bromophenol blue, 1000 mM NaCl, 4 M urea). Proteins were separated by 7.5% SDS-PAGE and transferred to a PVDF membrane (polyvinylidine difluoride membrane, Millipore). The membrane was blocked with TBS-T (20 mM Tris-HCl, pH 7.6, 136 mM NaCl, and 0.1% Tween-20) containing 3% BSA (Research Organics) for 1 hr, then incubated with the primary antibody solution at 4°C overnight. After washing with TBS-T, the membrane was incubated with horseradish peroxidase (HRP)-conjugated secondary antibody (Bio-Rad) for 1 hr at room temperature. Signals were detected by Chemiluminescence (Pierce).

## Protein purification

We expressed and purified Human Drp1 000 isoform (*Strack et al., 2013*) from *S. cerevisiae* and *E. coli*. Yeast-purified Drp1 was used for TIRF microscopy and negative staining electron microscopy. *E. coli*-purified Drp1 was used in high-speed co-sedimentation experiments and GTPase assays.

### Yeast purification

Drp1 was purified from yeast as previously described (*Koirala et al., 2013*). Both yeast strain (JSY9612, gentoype: MATa, can1 ade2, trp1, ura3, his3, leu2, pep4::HIS3, prb1::LEU2, bar1::HISG, lys2::GAL1/10-GAL4) and expression plasmids (pYSG-IBA167-hDRP1 and GYSG-IBA167-GFP-DRP1), were a gift from Janet Shaw and described previously (*Koirala et al., 2013*). We added an additional purification step of size-exclusion chromatography on Superdex200 (GE Biosciences). The Superdex200 buffer included 20 mM HEPES pH 7.5, 150 mM KCl, 2 mM MgCl$_2$, 1 mM DTT, 0.5 mM EDTA. The S200 Drp1 peak was concentrated using a 30,000 MWCO Amicon Ultra centrifugal filter (EMD Millipore UFC903024). Protein aliquots (25–100 μL) were frozen with liquid nitrogen and stored at -80°C. For the GFP-Drp1 construct, the GFP tag (A206K mutant) is located on the N-terminus of Drp1, and GFP-Drp1 was purified according to the same procedure.

### Bacterial purification

Human Drp1 isoform 000 was cloned with an N-terminal Strep-tag followed by an HRV3C protease site into the pET16b vector. Drp1 was expressed in One Shot BL21 Star (DE3) *E. coli* (Life Technologies C6010-03) by IPTG induction at 20°C for 16 hr. Cell pellets were resuspended in lysis buffer (100 mM Tris-Cl pH 8.0, 500 mM NaCl, 1 mM DTT, 1 mM EDTA, 2 μg/ml Leupeptin, 10 μg/ml Aprotinin, 2 μg/ml Pepstatin A, 2 mM Benzamidine, 1 μg/ml Calpain inhibitor I (ALLN), 1 μg/ml calpeptin). Cells were lysed using a high-pressure homogenizer (M-110L Microfluidizer Processor, Newton Massachusetts). The lysate was clarified by centrifugation at 40,000 rpm (Type 45 Ti rotor, Beckman) for 1 hr at 4°C. Avidin (20 μg/ml, Fisher Scientific PI-21128) was added, then the supernatant was loaded onto Strep-Tactin Superflow resin (IBA 2-1206-025) by gravity flow. The column was washed with 20-column volumes lysis buffer without protease inhibitors. To elute Drp1, 0.01 mg/ml HRV3C protease in lysis buffer without protease inhibitors was added for 16 hr at 4°C. The Strep-Tactin Superflow eluate was further purified by size exclusion chromatography, spin-concentrated, frozen liquid nitrogen and stored at -80°C, analogous to the yeast-expressed construct.

Rabbit skeletal muscle actin was purified from acetone powder as previously described (*Spudich and Watt, 1971*), and further purified by size exclusion chromatography on Superdex 75 (GE Biosciences). For TIRF microscopy and pyrene-actin experiments, actin was labeled with TAMRA NHS ester (Invitrogen C1171) or pyrene-iodoacetamide (ThermoFisher P-29) as described (*Gurel et al., 2014*). Actin was stored at 4°C in G-buffer (2 mM Tris, pH 8.0, 0.5 mM DTT, 0.2 mM ATP, 0.1 mM CaCl$_2$, and 0.01% NaN$_3$).

To prepare recombinant Mff cytoplasmic region, *E. coli* BL21 (DE3) (Invitrogen) were clonally grown overnight in SOB (carbenicillin at 100 mg/L) at 37°C while shaking. With an OD600 >1.5, protein was induced for 5.5 hr with isopropyl β-d-1-thiogalactoside, lactose and ampicillin added to final concentrations of 0.5 mM, 5 g/L and 50 mg/L, respectively. Cells were lysed in 50 mM Tris pH7.5, 150 mM NaCl, 2 mM benzamidine, and 0.1 mM PMSF, sonicated and centrifuged at 15,000 ×g for 1 hr at 4°C. The supernatant was discarded and protein was extracted from the pellet using a buffer containing 25 mM Hepes pH7.5, 50 mM NaCl, 8 M urea and 1 mM DTT. Protein was re-natured by sequential dialysis in 2-volumes of the same buffer without urea for 6 × 8 hr at 4°C, followed by size exclusion chromatography on Superdex200. Mff eluted as a single symmetrical peak. By velocity analytical ultracentrifugation (*Gurel et al., 2014*), Mff sedimented as a single species of 2.9 S particle in 25 mM Hepes pH 7.5, 150 mM NaCl, 1 mM DTT.

## High-speed co-sedimentation assay

Actin filaments were assembled from monomers (20 μM) for 1 hr at 23°C by addition of a 10x stock of polymerization buffer (500 mM NaCl, 10 mM MgCl$_2$, 10 mM EGTA, 100 mM imidazole pH 7.0) to a 1x final concentration. To maintain ionic strength across all samples, an actin blank was prepared in parallel using G-buffer in place of actin monomers, and used to dilute actin filaments as needed for each sample.

Drp1 was diluted to 10 µM in 150 mM NaCl, 1 mM MgCl$_2$, 1 mM EGTA, 10 mM Imidazole, then centrifuged at 100,000 rpm for 20 min at 4°C in a TLA-120 rotor (Beckman). The supernatant was stored on ice, and its protein concentration determined by Bradford assay (Bio-Rad 500-0006).

Drp1 (1.3 µM) was incubated with varying amounts of actin filaments (0.25–10 µM) for 1 hr at 23°C in a 200 µl volume. The final ionic strength was adjusted to 75 mM using NaCl. Following incubation, samples were centrifuged at 80,000 rpm for 20 min at 4°C in a TLA-100.1 rotor (Beckman). The supernatant was carefully removed, and 100 µl was mixed with SDS-PAGE sample buffer. Pellets were washed briefly and gently with 100 µl of 50 mM NaCl, 1 mM MgCl$_2$, 1 mM EGTA, 10 mM Imidazole pH 7.4, then resuspended in 100 µl SDS-PAGE sample buffer and resolved by SDS-PAGE. Gels were stained with Colloidal blue staining SDS-PAGE (Invitrogen LC6025), and band intensity was analyzed using ImageJ software.

## Total internal reflection (TIRF) microscopy

TAMRA-labeled actin (1 µM, 20% TAMRA labeled) was diluted in TIRF buffer (50 mM KCl, 1 mM MgCl$_2$, 1 mM EGTA, 10 mM Hepes pH 7.4, 100 mM DTT, 0.2 mM ATP, 15 mM Glucose, 0.5% Methyl Cellulose, 0.01 mg/ml catalase (Sigma C3515), 0.05 mg/ml glucose oxidase (Sigma G6125), 0.1% BSA) was polymerized for 10 min in glass flow chambers (see below), at which point indicated concentrations of GFP-Drp1 (diluted in TIRF buffer) was added. The filaments were visualized using an Olympus IX-83 inverted microscope equipped with a 4-channel CellTIRF attachment, appropriate lasers and driven by Metamorph for Olympus software. Simultaneous dual-color images were acquired every 1 s with TIRF objective (60x, 1.49 N.A.) and two Andor Zyla scMOS cameras with an Andor TuCam adapter.

Glass flow chambers were assembled using VWR microcover glasses (22 × 22 mm and 18 × 18 mm No 1.5) with double-stick scotch tape to hold 10 µl volume. Prior to chamber assembly, the cover glasses were washed with acetone (50 min), ethanol (10 min), water (1 min), and incubated in a solution containing a 1:2 ratio of 30% H$_2$O$_2$: concentrated H$_2$SO$_4$ for one hour. Cover glasses were then rinsed with water, 0.1 M KOH, and water again. Inert gas was used to completely dry cover glasses before silanization. Glasses were silanized overnight in a solution of 0.0025% dichlorodimethyl silane (Sigma 85126) in chloroform, then washed with methanol and dried with inert gas. Glasses were stored in a clean sealed container. Immediately before starting an assay, chambers were incubated with 1% Pluronic F127 (Sigma P2443) in BRB80 buffer (80 mM Pipes/KOH, pH 6.9, 1 mM EGTA, 1 mM MgCl$_2$) for 1 min, and then equilibrated with TIRF buffer.

## Negative staining electron microscopy

Drp1 was diluted in 50 mM KCl, 1 mM MgCl$_2$, 1 mM EGTA, 10 mM Hepes pH 7.4. To remove potential aggregates or small Drp1 nuclei, Drp1 was centrifuged at 100,000 rpm for 20 min at 4°C in a TLA-120 rotor (Beckman), and the supernatant was stored on ice for use. 4 µM actin was polymerized for 7 min in G-buffer plus 50 mM KCl, 1 mM MgCl$_2$, 1 mM EGTA, 10 mM Hepes pH 7.4. Drp1 was added to polymerized actin for a final concentration of 1.3 µM Drp1 and 2 µM actin. Samples (30 µl) were absorbed onto EM grids (Electron Microscopy Sciences, CF300-Cu) for 4 min, then blotted gently with filter paper. Grids were subsequently stained with 1% uranyl acetate solution for 1 min, and again blotted gently with filter paper. The prepared grids were imaged on a JEOL JEM 1010 transmission electron microscope operated at 100 keV acceleration. Images were collected using an XR-41B AMT digital camera and capture engine software (AMTV 540; Advanced Microscopy Techniques). Filament widths were quantified using ImageJ.

## GTPase assay

To remove ATP, actin monomers in G-buffer were incubated with Bio-Rad AG1-X2 100–200 mesh anion exchange resin (Dowex) (Bio-Rad, 1401241) rotating at 4°C for 5 min, followed by low-speed centrifugation to remove resin. Actin filaments (20 µM) were polymerized in 50 mM KCl, 1 mM MgCl$_2$, 1 mM EGTA, 10 mM Hepes pH 7.4 for 1 hr at 23°C. Drp1 was diluted in 150 mM KCl, 1 mM MgCl$_2$, 1 mM EGTA, 10 mM Hepes pH 7.4, then centrifuged to remove aggregates as described above.

Drp1 (1.3 µM) was mixed with actin filaments (1 µM), and the final ionic strength was adjusted to the equivalent of 75 mM KCl using 4 M KCl stock. Samples were incubated at 37°C for 5 min. At this

point, GTP was added to a final concentration of 250 µM to start reactions at 37°C. Reactions were quenched at various time points by mixing 20 µL of sample with 5 µL of 125 mM EDTA in a clear flat-bottomed 96-well plate (Greiner). Six time points were acquired for each condition. Inorganic phosphate was determined by addition of 150 µl of malachite green solution (1 mM Malachite green (Sigma Aldrich, 2290105–100 g), 10 mM ammonium molybdate ([Sigma Aldrich, A7302–100 g] in 1N HCl) to 25 µl quenched reactions. Absorbance at 650 nm was measured with a 96-well fluorescence plate reader (TECAN Infinite M1000, Mannedorf, Switzerland). GTP hydrolysis rates were determined by plotting phosphate concentration as a function of time.

## Pyrene-actin polymerization assays

Described in detail in *Gurel et al. (2014)*. Briefly, unlabeled and pyrene-labeled actin were mixed to a concentration of 6.67 µM (5% pyrene label) in G-buffer, then diluted to 6 µM by adding 0.1 volumes of 10 mM EGTA/1 mM MgCl2 for 2 min. Polymerization assays were initiated by diluting actin to 2 µM using 1.5x polymerization buffer containing the indicated concentrations of Drp1. Polymerization was monitored in a TECAN Infinite M1000 fluorescence plate reader at 365 nm excitation and 410 nm emission at 23°C. Time between reagent mixing and fluorescence recording was < 30 s.

## Acknowledgements

We thank Charles Barlowe, Tom Blanpied, Sai Divakaruni, Sajjan Koirala, Laura Lackner, Tina Maurot, James Moseley, Martin Pollak, Lori Schoenfeld, Janet Shaw and William Wickner. Supported by NIH GM069818, DK088826, and GM106000 to HNH, NS056244 and NS087908 to SS, an NSF Pre-doctoral Fellowship to ALH, and NIH S10OD010330 to the Norris Cotton Cancer Center Microscopy Facility.

## Additional information

### Funding

| Funder | Grant reference number | Author |
|---|---|---|
| National Institutes of Health | GM069818 | Henry N Higgs |
| National Institutes of Health | GM106000 | Henry N Higgs |
| National Institutes of Health | DK088826 | Henry N Higgs |
| National Institutes of Health | NS056244 | Stefan Strack |
| National Institutes of Health | NS087908 | Stefan Strack |
| National Science Foundation | | Anna L Hatch |

The funders had no role in study design, data collection and interpretation, or the decision to submit the work for publication.

### Author contributions

W-KJ, ALH, Conception and design, Acquisition of data, Analysis and interpretation of data, Drafting or revising the article; RAM, Ronald Merrill made a key reagent (Mff) essential to experiments conducted for the revision. His insight into the design and productive of this reagent was beyond that of mere technical assistance., Acquisition of data, Contributed unpublished essential data or reagents; SS, Drafting or revising the article, Contributed unpublished essential data or reagents; HNH, Conception and design, Analysis and interpretation of data, Drafting or revising the article

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
