## [Decision Letter]

Thank you for submitting your work entitled "Actin filaments target the maturation of Drp1 oligomers to mitochondrial fission sites" for consideration by *eLife*. Your article has been favorably assessed by Randy Schekman (Senior Editor) and three peer reviewers, one of whom, Pekka Lappalainen, is a member of our Board of Reviewing Editors. The other two reviewers were Liza Pon and Aurelien Roux.

The reviewers have discussed the reviews with one another and the Reviewing editor has drafted this decision to help you prepare a revised submission.

Drp1 is a dynamin-like GTPase, which promotes mitochondrial fission. However, the mechanisms underlying Drp1 association, oligomerization and dynamics on mitochondrial membrane are incompletely understood. Here, Ji et al., demonstrate that mitochondrially-bound Drp1 oligomers are motile and undergo merging events in order to mature into stable oligomers that are associated with mitochondrial fission. They also show that only a fraction of mitochondrially-associated Drp1 oligomers catalyze fission and that there is typically a long lag-period between the assembly/maturation of Drp1 puncta and the actual fission event. Finally, they provide evidence that actin filaments increase Drp1 accumulation and mitochondrial fission, and that Drp1 directly interacts with actin.

The majority of experiments presented are convincing and of very good technical quality, and this interesting study extends our understanding of the mechanism underlying mitochondrial fission and provide evidence for a novel role of the actin cytoskeleton in recruiting Drp1 to the organelle. However, certain aspects of the study are slightly premature and at a somewhat descriptive level. Thus, few additional experiments geared towards understanding the mechanism and function of the peculiar behavior of Drp1 would significantly strengthen the manuscript.

Essential revisions:

1) Actin-binding (Figure 8) of Drp1 is an interesting observation, but whether the direct association between Drp1 and F-actin has any relevance in cells remains unclear. At minimum, the authors should examine whether Drp1 that is bound to a cardiolipin-rich membrane is capable of interacting with F-actin. This could be tested e.g. by examining (by fluorescence microscopy or by co-flotation assays) whether cardiolipin-rich vesicle/liposome -associated Drp1 (see e.g. Macdonald et al., MBoC, 2014) is capable of recruiting F-actin on the membrane in vitro. Alternatively, the authors could apply mutagenesis to reveal whether the binding sites for F-actin and membrane overlap on the surface of Drp1.

2) Previous studies from this group revealed that silencing of INF2 or Myosin II results in decrease in localization of Drp1 puncta to mitochondria. At the time, they had not resolved high and low threshold puncta. Nonetheless, this suggests that the actin-dependent recruitment of Drp1 to mitochondria is a general phenomena and not a consequence of ionomycin treatment. However, all studies on INF2, Myosin II and actin were performed in the presence of ionomycin. The manuscript would benefit from analysis of the effect of LatA treatment and of silencing INF2 or myosin II on recruitment of high and low threshold Drp1 puncta to mitochondria in cells that are not treated with ionomycin.

3) The process where short oligomers are assembled into a productive, complete ring of Drp1 at the mitochondria may actually have a function. Is it possible this function could be to estimate the total length of the mitochondria, in order to fission the longest mitochondria in priority? Have the authors measured the probability of fission as a function of mitochondrial length? Such analysis may be very informative. Also, it would be better to present the fission data as a percentage of rings undergoing fission during the 10 min observation period, as this is more intuitive to understand than the fission efficiency presented in Figure 3 (and to eventually see that perturbing the actin cytoskeleton changes this correlation). Thus, few additional assays, data analysis and/or discussion concerning the possible cellular function of the peculiar assembly mechanism of Drp1 would significantly strengthen the manuscript (please see also 'minor points by reviewer #2' for further suggestions concerning possible additional experiments to test this hypothesis).

4) The authors used mitochondria-targeted BFP and time-lapse imaging to visualize mitochondria association of Drp1 with the organelle and mitochondrial fission. Extensive UV illumination required to excite BFP can be damaging and induce mitochondrial fission. The manuscript would benefit from studies to determine whether the BFP-imaging conditions used affect mitochondria and cells.

Minor points:

*Reviewer #1*:

1) In the Results, K_d_ value for Drp1 – actin interaction should be 0.8 μM (not 0.8 mM as stated in the text).

2) Why are the average mitochondrial diameters different for 'non-productive Drp1 bound mitochondria' between Figure 4 and Figure 4? This should be explained.

3) In the co-sedimentation assay (Figure 8) the binding of 1,3 μM Drp1 reaches a plateau at 0,5 μM (i.e. more than 50% of the Drp1 is unable to co-sediment with F-actin). Does this mean that majority of Drp1 used in the assay was inactive?

4) Does Drp1 bundle F-actin? Can the authors exclude the possibility that the thick actin filaments in Figure 8 do not result from filament bundling (instead of Drp1-decoration)?

*Reviewer #2*:

1) It seems intuitive that fission rate should be proportional to the length of the mitochondria, the longest ones being chopped into smaller ones. Have the authors measure the probability of fission as a function of mitochondrial length?

2) I further wonder if longer mitochondria will then be covered by more punctae of DRP1, even though their density (number of punctae divided by length) could be constant. Have the authors measured the density of punctae as a function of mitochondria length?

3) If point 2 is correct, longer mitochondria may thus have more punctae, which then will create more rings to fission more longer mitochondria. Have the authors measured the number of productive rings (undergoing fission) as a function of the length of mitochondria?

4) If the proposed points above are correct, than one can understand the function of the actin-dependent mechanism of assembly shown by the authors. The role of the actin will then be to gather together oligomer evenly distributed at the surface of the mitochondria into productive rings at a fairly constant distance between productive rings: it means that the distribution of distances between productive (leading to fission) rings could be fairly broad, but with a peak. Have the authors measured the distribution of distances between fission points or productive rings?

5) In this case, one expects the distance between the rings to be strongly affected by actin polymerization dynamics. A simple expectation could be that fast turn over could make closer rings and lower dynamics rings more spread apart, or even absent, which is what the authors have observed. Have the authors tried to overexpress cortactin (promotes actin polymerization) and/or cofilin (enhance turnover of actin) to see the effects of these molecules on the distance between productive rings?

6) How did the authors choose the 30% high threshold? It sounds fairly arbitrary, is there any logic behind it?

7) I found a striking difference between several fission events reported in this manuscript: some of them are left with DRP1 on one of the 2 tips of the mitochondria (tips generated by the fission), and others seem to have DRP1 at both tips. In this last case, they usually are of very different intensity. I was thus wondering if the authors had noticed this, and if most of the case were only one tip decorated with Drp1 or the 2 tips were always decorated.

8) As said above, the fission efficiency parameter is fairly non-intuitive, a percentage of dots reaching fission within the 10min of observation would be better, I think.

9) A small fraction of the low threshold dots still reach fission, so I was wondering if in doing so, their intensity was growing to the high threshold category or would stay constant.

10) The authors have made a stable cell line expressing DRP1-GFP representing 44% of the total amount of DRP found in those cells. I was wondering if the authors could check further the functionality of the GFP construct by doing a siRNA against the endogenous form of DRP1 (which could be feasible if the authors have used a cDNA construct for doing the stable cell line), and show that the fission of mitochondria is still close to normality?

11) 10 min of observation looks short in comparison to the duration of assembly and fission of mitochondria. I understand that the imaging of these events is difficult, but could it be possible to image them for much longer, in order to have a proper statistics (many rings may end up in being productive, but just not in the time window selected by the authors).

12) Kymographs of single dots forming and fissioning could help in visualizing better the kinetics of assembly and fission. Not absolutely required.

*Reviewer #3*:

1) The scale on the Y-axis needs to extend from 0 in Figure 3, Figure 4, Figure 6, and Figure 7 and many of the supplemental figures. If the authors want to emphasize the differences observed under different experimental conditions, they can show a full scale and an insert showing a zoom in on areas of interest.

2) The figure legend for Figure 7 does not provide sufficient information for the reader to understand what was done. The graphs show fluorescence intensity of Drp1 and actin under conditions of ionomycin treatment. Is this the fluorescence intensity of Drp1 or actin that is associated with mitochondria? The figure legend and Y-axis labels should be revised to clarify this issue.

3) The figure legend for Figure 8 does not provide sufficient information for the reader to understand what was done. Is actin maintained as filaments (e.g. by phalloidin) in the experiments for Figure 8 and its associated supplemental figures? If not, is the concentration of F- and F-actin known under their experimental conditions? The figure legend should be revised for clarity.

4) In Figure 8, the K_d_ of Drp1 binding to F-actin is 0.8 µM and in the text, the K_d_ is 0.8 mM. Based on the data shown, the reference to mM is a typo, which should be corrected.

5) Based on the negative staining results shown in Figure 8, purified Drp1 does not change the width of F-actin. Instead, it results in bundling of actin filaments. The text should be modified accordingly.

6) The figure legend for Figure 8—figure supplement 1 does not provide sufficient information for the reader to understand what was done and what is shown. Do the lanes labeled "standards" show the total amount of Drp1 and actin in the binding mixture without centrifugation? Are the lanes in the labeled Drp1, µM and actin µM the pellets from the ultracentrifugation? The legend needs to be revised for clarity.

---

## [Author Response]

*[…] The majority of experiments presented are convincing and of very good technical quality, and this interesting study extends our understanding of the mechanism underlying mitochondrial fission and provide evidence for a novel role of the actin cytoskeleton in recruiting Drp1 to the organelle. However, certain aspects of the study are slightly premature and at a somewhat descriptive level. Thus, few additional experiments geared towards understanding the mechanism and function of the peculiar behavior of Drp1 would significantly strengthen the manuscript.*

We agree that there is much to do in obtaining a full mechanistic model of this process. However, we feel that our findings make an important mechanistic advance, in showing that the assembly of productive Drp1 oligomers on mitochondria is through alteration of an equilibrium process that is continuously occurring on mitochondria. Drp1 is constantly binding to and releasing from mitochondria, as well as in equilibrium between oligomeric states both on and off mitochondria, even in the absence of signals for fission. Fission signals (in this case, actin filaments near the fission site) serve to shift this equilibrium toward productive Drp1 oligomerization at fission sites. Our model contrasts with previous models depicting that Drp1 is recruited directly from the cytosol to fission sites in response to fission signals.

*Essential revisions:*

*1) Actin-binding (Figure 8) of Drp1 is an interesting observation, but whether the direct association between Drp1 and F-actin has any relevance in cells remains unclear. At minimum, the authors should examine whether Drp1 that is bound to a cardiolipin-rich membrane is capable of interacting with F-actin. This could be tested e.g. by examining (by fluorescence microscopy or by co-flotation assays) whether cardiolipin-rich vesicle/liposome -associated Drp1 (see e.g. Macdonald et al., MBoC, 2014) is capable of recruiting F-actin on the membrane in vitro. Alternatively, the authors could apply mutagenesis to reveal whether the binding sites for F-actin and membrane overlap on the surface of Drp1.*

This is, indeed, an important question that we have pursued in a somewhat different manner than that suggested. We chose to examine the combined effects of actin filaments and Mff, a Drp1 receptor on the OMM. We chose Mff for the following reason: this protein has been shown to be important for mitochondrial fission by multiple groups (Gandre-Babbe.et al., 2008; Otera et al., 2010; Loson et al., 2013; Liu et al. 2015). We now show evidence that actin filaments and Mff (the cytoplasmic portion) synergize to increase Drp1’s GTPase activity (Figure 9 and Results). These are very exciting results that lead to many new mechanistic hypotheses for us (included in Discussion). Although cardiolipin is an intriguing fission factor, we feel that it could be an “alternate” co-activator to actin filaments (both being anionic multivalent surfaces). Our future experiments will focus on mechanistically dissecting the combined effects of actin filaments, cardiolipin and Mff (as well as other OMM receptors) on Drp1 activity. Resolution of this question will require extensive experimentation that we feel is beyond the scope of this manuscript, so we hope that the new data we present are sufficient to provide evidence for the potential physiological significance of the actin/Drp1 interaction.

*2) Previous studies from this group revealed that silencing of INF2 or Myosin II results in decrease in localization of Drp1 puncta to mitochondria. At the time, they had not resolved high and low threshold puncta. Nonetheless, this suggests that the actin-dependent recruitment of Drp1 to mitochondria is a general phenomena and not a consequence of ionomycin treatment. However, all studies on INF2, Myosin II and actin were performed in the presence of ionomycin. The manuscript would benefit from analysis of the effect of LatA treatment and of silencing INF2 or myosin II on recruitment of high and low threshold Drp1 puncta to mitochondria in cells that are not treated with ionomycin.*

We have now quantified the effects of INF2 and myosin IIA suppression, as well as LatA treatment, on mitochondrially-bound Drp1 puncta (both total puncta and high threshold puncta), and find significant reductions for all these treatments. These results are now depicted in Figure 6—figure supplement 1, and described in the Results section.

*3) The process where short oligomers are assembled into a productive, complete ring of Drp1 at the mitochondria may actually have a function. Is it possible this function could be to estimate the total length of the mitochondria, in order to fission the longest mitochondria in priority? Have the authors measured the probability of fission as a function of mitochondrial length? Such analysis may be very informative.*

We now supply data showing that the mitochondria undergoing fission tend to be the longer ones (Figure 3—figure supplement 1 and paragraph seven, Results). There are several issues that complicate this analysis. First, most mitochondria are in a network in U2OS cells, which is branched. This network often extends from the periphery to the peri-nuclear region, and it is often difficult to find both ends. Second, the frequency of mitochondrial fission is relatively low and our viewing time is limited (see reviewer 2 comments), making it difficult to conduct this analysis on a large number of events at present. Given these difficulties, we analyze fission events on mitochondria for which we can measure lengths, and find the mitochondria undergoing fission to be generally longer than those not undergoing fission. We also analyze in more detail the number of Drp1 puncta as a function of mitochondrial length, in response to reviewer 2 comment 2, and find that Drp1 punctum number increases linearly with mitochondrial length (Figure 1—figure supplement 1). Thus, Drp1 punctum density does not vary with mitochondrial length. Finally, we examine the relationship between mitochondrial length and the “lag time” (between stable Drp1 oligomer assembly and fission) and find no apparent correlation (Figure 3, paragraph six, Results). We think that the aggregate of the information suggests that the reason that fission occurs more readily on longer mitochondria is that there are more Drp1 puncta overall (but not higher density), providing a higher probability that one will become fission-competent. In the Discussion, we add a statement that the issue of mitochondrial branching, and its influence on mitochondrial fission, is a fascinating question still to be addressed (paragraph three, Discussion).

*Also, it would be better to present the fission data as a percentage of rings undergoing fission during the 10 min observation period, as this is more intuitive to understand than the fission efficiency presented in Figure 3 (and to eventually see that perturbing the actin cytoskeleton changes this correlation).*

We have altered Figure 3 and modified the Results to reflect the reviewers’ comment.

*Thus, few additional assays, data analysis and/or discussion concerning the possible cellular function of the peculiar assembly mechanism of Drp1 would significantly strengthen the manuscript (please see also 'minor points by reviewer #2' for further suggestions concerning possible additional experiments to test this hypothesis).*

We have altered our Discussion to reflect these comments (paragraph three), but we feel that the assembly mechanism is not necessarily peculiar. Our hypothesis is that fission signals such as actin filaments push the equilibrium of Drp1 oligomerization on mitochondria, driving maturation. Our evidence throughout the paper is consistent with this possibility. This mechanism of driving a reaction through changing an equilibrium process is used to regulate a large number of biological processes (metabolism, actin polymerization, many others) so we do not regard it as particularly bizarre.

*4) The authors used mitochondria-targeted BFP and time-lapse imaging to visualize mitochondria association of Drp1 with the organelle and mitochondrial fission. Extensive UV illumination required to excite BFP can be damaging and induce mitochondrial fission. The manuscript would benefit from studies to determine whether the BFP-imaging conditions used affect mitochondria and cells.*

We agree that this is a concern, and have measured fission rates in the absence and presence of ionomycin using either our mitoRed or mitoBFP markers. In this experiment, we imaged for 25 min, which is longer than we have for the other experiments in this manuscript (see response to a reviewer 2 comment). We found similar fission rates for either mitochondrial matrix probe.

These results are illustrated in Figure 6—figure supplement 3 and discussed in the Results.

*Minor points:* Reviewer #1:

We appreciate the reviewer’s comments, which enabled us to bring up two important points about Drp1 biochemical properties.

1) In the Results, K_d_ value for Drp1 – actin interaction should be 0.8 μM (not 0.8 mM as stated in the text).

Thank you for catching this. We have now corrected it.

*2) Why are the average mitochondrial diameters different for 'non-productive Drp1 bound mitochondria' between Figure 4 and Figure 4? This should be explained.*

We are sorry for the confusion on this issue, which was due to poor explanation in the text and figures. Figure 4 examines mitochondrial diameter (through Tom20) and Figure 4 examines Drp1 punctum diameter. We have improved the figure labeling to clarify this issue.

*3) In the co-sedimentation assay (Figure 8) the binding of 1,3 μM Drp1 reaches a plateau at 0,5 μM (i.e. more than 50% of the Drp1 is unable to co-sediment with F-actin). Does this mean that majority of Drp1 used in the assay was inactive?*

The reviewer was very astute in noticing this. We know that the vast majority of our Drp1 is active, since we get close to 100% pelleting in the presence of non- hydrolyzable GTP analogue. We now include these data in Figure 8—figure supplement 1 and discuss this phenomenon in the Results. We have been fascinated by this issue, and are investigating it in depth. It is due to a combination of varying Drp1 oligomeric state (each state with varying affinity for actin filaments) and a more complicated relationship between Drp1- Drp1 interactions and off-rate of Drp1 from actin filaments. We would ask to be allowed to address this mechanism in more depth in a subsequent publication, because it merits a more detailed analysis than we can do in this manuscript.

4) Does Drp1 bundle F-actin? Can the authors exclude the possibility that the thick actin filaments in Figure 8 do not result from filament bundling (instead of Drp1-decoration)?

This is a great observation. Indeed, in our EMs we cannot distinguish between Drp1 bound to single filaments versus Drp1-bundled filaments. Drp1 can induce filament bundling, and we now supply evidence for this (Figure 8; Results). As with many facets of Drp1 biochemistry, this bundling is not straightforward (depicted in 8F), and we would prefer to show initial evidence for bundling here, followed by more detailed examination in a future publication.

Reviewer #2:

We appreciate the thoughtful comments by this reviewer, which challenged us and made us think about the process in a different way. Ultimately, our responses to his/her specific comments are somewhat ‘negative’ in that we were not able to obtain clear evidence in support of their idea.

However, the reviewer’s comments bring up issues of the relationship between mitochondrial fission and the mitochondrial network, including many fascinating issues with mitochondrial branches: how are branches made, and how do they affect fission? We hope to address these in the future, but feel it appropriate to mention these only in passing in the Discussion.

*1) It seems intuitive that fission rate should be proportional to the length of the mitochondria, the longest ones being chopped into smaller ones. Have the authors measure the probability of fission as a function of mitochondrial length?*

We have analyzed the lengths of filaments undergoing fission, as well as those not experiencing fission during the viewing period, and find that longer filaments tend to be the ones undergoing fission. This is now Figure 3—figure supplement 1. These analyses are complicated by the fact that it is difficult to define length, given the mitochondrial network that exists.

*2) I further wonder if longer mitochondria will then be covered by more punctae of DRP1, even though their density (number of punctae divided by length) could be constant. Have the authors measured the density of punctae as a function of mitochondria length?*

We have now measured Drp1 puncta number as a function of mitochondrial length, and find that, as suspected by the reviewer, the relationship is roughly linear for both total puncta and for high threshold puncta. These data are now in Figure 1—figure supplement 1, and are described in paragraph three of the Results.

*3) If point 2 is correct, longer mitochondria may thus have more punctae, which then will create more rings to fission more longer mitochondria. Have the authors measured the number of productive rings (undergoing fission) as a function of the length of mitochondria?*

We have found that longer mitochondria are more likely to undergo fission in our viewing period (response to major comment 3 and to reviewer 2 comment 1). This essentially corresponds to productive rings (since all fission events correspond to high-threshold Drp1 rings). However, the fact that the density of high-threshold puncta does not increase with mitochondrial length (response to reviewer 2 comment 2) suggests that maturation is not favored on longer mitochondria. The conversion phase does not seem to be affected by mitochondrial length, if one can use lag time as a measure of conversion (Figure 3).

*4) If the proposed points above are correct, than one can understand the function of the actin-dependent mechanism of assembly shown by the authors. The role of the actin will then be to gather together oligomer evenly distributed at the surface of the mitochondria into productive rings at a fairly constant distance between productive rings: it means that the distribution of distances between productive (leading to fission) rings could be fairly broad, but with a peak. Have the authors measured the distribution of distances between fission points or productive rings?*

It is difficult to measure the distance between productive rings on mitochondria, because the majority of fission events occur in isolation. Further response to this comment is given in comment 5.

*5) In this case, one expects the distance between the rings to be strongly affected by actin polymerization dynamics. A simple expectation could be that fast turn over could make closer rings and lower dynamics rings more spread apart, or even absent, which is what the authors have observed. Have the authors tried to overexpress cortactin (promotes actin polymerization) and/or cofilin (enhance turnover of actin) to see the effects of these molecules on the distance between productive rings?*

Our hypothesis is that actin filaments do not represent a “steady-state” signal to stimulate Drp1 maturation, but are activated to assemble at specific sites (in other words, a “fission signal”).

These site-specific signals serve to shift Drp1’s equilibrium towards assembly of productive oligomers. We have re-written the first paragraph of the Discussion to state this more clearly.

In this context, shifting global actin polymerization could have effects on mitochondrial fission, by changing the availability of polymerizable actin monomers for example. The fact that INF2 is ER-bound (and that ER interacts with mitochondria at fission sites), and our previous localization studies of activated myosin II suggest that these influence mitochondrial fission specifically. We are interested in the potential roles of cortactin and cofilin in this process, especially due to their published links to fission (referenced in the manuscript), but feel that significant experimentation is required to deconvolve their global roles from mitochondria-specific roles.

*6) How did the authors choose the 30% high threshold? It sounds fairly arbitrary, is there any logic behind it?*

We did not have a specific logic in choosing the 30% cut-off point, since the distribution in punctum intensity is fairly continuous. The high-threshold designation allows us to distinguish the more mature puncta from total puncta, which allows us a means of assessing maturation. We now include the word “arbitrarily” in paragraph two of the Results section.

*7) I found a striking difference between several fission events reported in this manuscript: some of them are left with DRP1 on one of the 2 tips of the mitochondria (tips generated by the fission), and others seem to have DRP1 at both tips. In this last case, they usually are of very different intensity. I was thus wondering if the authors had noticed this, and if most of the case were only one tip decorated with Drp1 or the 2 tips were always decorated.*

We thank the reviewer for this observation, and now show examples of Drp1 remaining at newly created mitochondrial ends. From our analysis, 30% of the time the Drp1 remains only at one of the two new ends. We now show this in Figure 3—figure supplement 1, and describe it in the Results.

*8) As said above, the fission efficiency parameter is fairly non-intuitive, a percentage of dots reaching fission within the 10min of observation would be better, I think.*

We have altered Figure 3 to reflect the reviewer’s comment.

*9) A small fraction of the low threshold dots still reach fission, so I was wondering if in doing so, their intensity was growing to the high threshold category or would stay constant.*

We now realize that our “low threshold” classification is a very poor use of words, since it refers to all Drp1 puncta with intensities over that of the diffuse background. We have changed “low threshold puncta” to “total puncta” throughout the manuscript.

*10) The authors have made a stable cell line expressing DRP1-GFP representing 44% of the total amount of DRP found in those cells. I was wondering if the authors could check further the functionality of the GFP construct by doing a siRNA against the endogenous form of DRP1 (which could be feasible if the authors have used a cDNA construct for doing the stable cell line), and show that the fission of mitochondria is still close to normality?*

We were also concerned about the ability of GFP-Drp1 to replace endogenous Drp1. Our original intent was to develop a stable cell line that both suppressed endogenous Drp1 and expressed GFP-Drp1 at near endogenous levels (hence use of the combined shRNA/re- expression construct). The clone that best suited our purposes was one that still had appreciable endogenous Drp1 expression, but whose GFP-Drp1 expression was relatively low (resulting in a <twofold increase in overall Drp1).

As partial response to this comment, we have analyzed a different clone isolated during production of our stable cell line. This clone effectively suppresses endogenous Drp1 to undetectable levels, while expressing GFP-Drp1 to levels ~threefold higher than Drp1 levels in control cells. This clone displays similar mitochondrial fission rates to those of control cells (both unstimulated and ionomycin-stimulated). We now show these data in Figure 1—figure supplement 1, and describe them in paragraph one, Results.

*11) 10 minutes of observation looks short in comparison to the duration of assembly and fission of mitochondria. I understand that the imaging of these events is difficult, but could it be possible to image them for much longer, in order to have a proper statistics (many rings may end up in being productive, but just not in the time window selected by the authors).*

We agree with the reviewer that, ideally, imaging would be conducted over a longer time period to capture more fission events and get a better gauge of Drp1 ring productivity. We used 10 minutes, though, because we found it necessary to acquire images as rapidly as possible, since fission events can happen quickly, followed by rapid mitochondrial translocation (causing an under- estimation of fission). Given the frequency of image acquisition, we limited total exposure time to reduce the potential for imaging-induced artifacts and photo-bleaching. In our revised manuscript, however, we have conducted the analysis of two experiments for > 20 minutes (Figure 1—figure supplement 1; and Figure 6—figure supplement 3) and have found no significant difference in fission rate from our 10 minute observation, which suggests that we are not experiencing imaging artifacts to a large degree.

*12) Kymographs of single dots forming and fissioning could help in visualizing better the kinetics of assembly and fission. Not absolutely required.*

We appreciate the reviewer’s suggestion, and have attempted kymograph analysis of Drp1 puncta before and during fission. Unfortunately, we found this analysis was complicated by the dynamics of the mitochondria themselves, which in our hands made the analysis too difficult to provide clear kinetic patterns for Drp1.

Reviewer #3:

We appreciate the reviewer’s comments, which led us to clarify a number of places that were confusing in the manuscript, and to address the actin bundling issue.

*1) The scale on the Y-axis needs to extend from 0 in Figure 3, Figure 4, Figure 6, and Figure 7 and many of the supplemental figures. If the authors want to emphasize the differences observed under different experimental conditions, they can show a full scale and an insert showing a zoom in on areas of interest.*

We have changed the Y axis to extend from 0 in Figure 3 and Figure 4. For Figure 6 and Figure 7, we feel that extending the Y axis to 0 in the main figure (and supplying a zoomed inset) is confusing and detracts from interpretation of the data. By limiting the Y axis, we are not trying to make the results appear more significant than they are, we are simply trying to display the data in the clearest manner possible. These are not large% changes in the Drp1 punctum or actin filament signals, but they are robust (as denoted by the error bars for each time point).

*2) The figure legend for Figure 7 does not provide sufficient information for the reader to understand what was done. The graphs show fluorescence intensity of Drp1 and actin under conditions of ionomycin treatment. Is this the fluorescence intensity of Drp1 or actin that is associated with mitochondria? The figure legend and Y-axis labels should be revised to clarify this issue.*

We have described the experiments better in the new figure legend. Briefly, both the Drp1 and actin signals represent fluorescence intensities above the diffuse cytosolic background, which represents Drp1 puncta and actin filaments, respectively. In both cases, these are not mitochondrially-localized Drp1 or actin (Drp1 is a whole cell evaluation, actin is evaluation of specific regions of interest).

*3) The figure legend for Figure 8 does not provide sufficient information for the reader to understand what was done. Is actin maintained as filaments (e.g. by phalloidin) in the experiments for Figure 8 and its associated supplemental figures? If not, is the concentration of F- and F-actin known under their experimental conditions? The figure legend should be revised for clarity.*

We have revised the legend to clarify this issue. No phalloidin is used in these assays, so that the concentration of actin filaments is the total actin concentration minus the barbed end critical concentration (0.1 μM). We know that Drp1 does not significantly change the kinetics of actin polymerization or the barbed end critical concentration, as determined by pyrene-actin polymerization assay (Figure 8—figure supplement 1).

*4) In Figure 8, the K_d_ of Drp1 binding to F-actin is 0.8 µM and in the text, the K_d_ is 0.8 mM. Based on the data shown, the reference to mM is a typo, which should be corrected.*

We regret this mistake and have corrected it.

*5) Based on the negative staining results shown in Figure 8, purified Drp1 does not change the width of F-actin. Instead, it results in bundling of actin filaments. The text should be modified accordingly.*

We have modified this section of the manuscript, in response to comments from both reviewer 1 and reviewer 3. In fact, we cannot distinguish if the negative staining represents an actin bundle or a single actin filament decorated by Drp1. We now supply evidence that Drp1 can cause actin filament bundling.

These results are shown in Figure 8, and described in the Results.

*6) The figure legend for Figure 8—figure supplement 1 does not provide sufficient information for the reader to understand what was done and what is shown. Do the lanes labeled "standards" show the total amount of Drp1 and actin in the binding mixture without centrifugation? Are the lanes in the labeled Drp1, µM and actin µM the pellets from the ultracentrifugation? The legend needs to be revised for clarity.*

We have now clarified that the “Standards” do indeed show the total amount of Drp1 and actin present in the assay prior to centrifugation. The other lanes are the pellet fractions and are now labeled as such.